# Spatial-aware decision-making with ring attractors in Reinforcement Learning systems

## Abstract

This paper explores the integration of ring attractors, a mathematical model inspired by neural circuit dynamics, into the reinforcement learning (RL) action selection process. Ring attractors, as specialized brain-inspired structures that encode spatial information and uncertainty, offer a biologically plausible mechanism to improve learning speed and predictive performance. They do so by explicitly encoding the action space, facilitating the organization of neural activity, and enabling the distribution of spatial representations across the neural network in the context of deep RL. The application of ring attractors in the RL action selection process involves mapping actions to specific locations on the ring and decoding the selected action based on neural activity. We investigate the application of ring attractors by both building them as exogenous models and integrating them as part of a Deep Learning policy algorithm. Our results show a significant improvement in state-of-the-art models for the Atari 100k benchmark. Notably, our integrated approach improves the performance of state-of-the-art models by half, representing a 53% increase over selected baselines.

## 1 Introduction

This paper addresses the challenge of efficient action selection in reinforcement learning (RL), particularly in environments with spatial structures. Our primary contribution is the novel integration of ring attractors, theoretically proposed by Zhang (1996) and discovered by Kim et al. (2017), a neural circuit model from neuroscience, into the RL framework. This approach improves spatial awareness in action selection and provides a mechanism for uncertainty-aware decision-making in RL, leading to more accurate and efficient learning in complex environments. Ring attractors offer a unique framework to continuously and stably represent spatial information (Sun et al., 2020). In a ring attractor network, neurons interconnect circularly, forming a loop with tuned connections (Blair et al., 2014). This configuration allows for robust and localized activation patterns, maintaining accurate spatial representations even with noise or perturbations. Applying ring attractors to the selection of RL actions involves mapping actions to specific ring locations and decoding the selected action based on neural activity. This spatial embedding proves advantageous for continuous action spaces, particularly in tasks such as robotic control and navigation (Rivero-Ortega et al., 2023). Ring attractors improve decision-making by exploiting spatial relations between actions, contributing to informed transitions between actions in sequential decision-making tasks in RL.

In what follows, we summarize our contributions. Briefly, our contributions include a novel approach to RL policies based on ring attractors, the inclusion of uncertainty-aware capabilities in our RL systems, and the development of Deep Learning (DL) modules for RL with ring attractors.

**Integration of ring attractors into RL policies and spatial encoding for action selection**. We propose a novel approach for incorporating ring attractors, a neural structure use for motor control and cognition, into RL as a lightweight, efficient and robust decision-making structure. The circular structure of ring attractors allows the model to represent spatial information and relations between actions. This spatial awareness significantly speeds up the learning rate of the RL agent. The relevant methodology and experiments can be found in Sections 3.1.2 and 4.1, respectively.

**Uncertainty-aware RL**. Ring attractors can encode uncertainty estimation to drive the action selection process. This paper utilizes Bayesian uncertainty estimation to influence the policy. The relevant methodology and experiments can be found in Sections 3.1.3 and 4.1, respectively.

**DL module for ring attractors**. We develop a reusable DL module based on recurrent neural networks that integrates ring attractors into DL-based RL agents. Additionally, this enables the adoption of our ring attractor approach across different RL models and tasks in various domains of application. The relevant methodology and experiments can be found in Sections 3.2 and 4.2, respectively.

## 2 RELATED WORK

The integration of ring attractors into RL systems brings together neuroscience-inspired models and advanced machine learning techniques. Here, we review the literature on the key areas that form the foundation of our RL research: spatial awareness in RL, biologically inspired reinforcement learning approaches, and uncertainty quantification methods.

### 2.1 SPATIAL AWARENESS IN REINFORCEMENT LEARNING

Incorporating spatial awareness into RL systems has improved performance on tasks with inherent spatial structure. Regarding relational RL, Zambaldi et al. (2019) introduced an approach using attention mechanisms to reason about spatial relations between entities in an environment. This method demonstrated improved sample efficiency and generalization in tasks that require spatial reasoning. On the topic of navigation, Mirowski et al. (2017) developed a deep RL agent capable of navigating complex city environments using street-level imagery. Their approach incorporated auxiliary tasks, such as depth prediction and loop closure detection. Concerning explicit spatial representations, Gupta et al. (2017) proposed a cognitive mapping and planning approach for visual navigation, combining spatial memory with a differentiable neural planner. Similarly, Bapst et al. (2019) introduced a relational deep RL framework using graph neural networks to capture spatial relations between objects. Although these approaches demonstrate the importance of spatial awareness in RL, they often lack the biological plausibility found in neural circuits.

### 2.2 BIOLOGICALLY INSPIRED MACHINE INTELLIGENCE

Biologically inspired approaches to RL seek to leverage insights from neuroscience to improve the efficiency, adaptability, and interpretability of RL algorithms. These methods often draw upon neural circuit dynamics and cognitive processes observed in biological systems. The work presented in (Banino et al., 2018) demonstrated that incorporating grid-like representations, inspired by mammalian grid cells, into RL agents improved performance on navigation tasks. Their work showed that these biologically inspired representations emerged naturally in agents trained on navigation tasks and transfer well to new environments. Similarly, Cueva & Wei (2018) showed that recurrent neural networks trained on navigation tasks naturally developed grid-like representations, suggesting a deep connection between biological and artificial navigation systems. Singh et al. (2023) demonstrated how RL agents naturally develop insect-like behaviors and neural dynamics when solving complex spatial navigation tasks. Wang et al. (2018) proposed a biologically inspired meta-RL algorithm that mimics the function of the prefrontal cortex and dopamine-based neuromodulation. Their approach demonstrated rapid learning and adaptation to new tasks, similar to the flexibility observed in biological learning systems.

### 2.3 UNCERTAINTY QUANTIFICATION

Regarding exploration strategies, Osband et al. (2016) introduced bootstrapped deep Q-networks (DQNs), addressing exploration by leveraging uncertainty in Q-value estimates by training multiple DQNs with shared parameters. Building on this theme, Burda et al. (2018) proposed random network distillation (RND), measuring uncertainty by comparing predictions between a target network and a randomly initialized network. For efficient uncertainty quantification, Durasov et al. (2020) and Bykovets et al. (2022) presented a novel 'masksemble' approach, applying masks across the input batch during the forward pass to generate diverse predictions. Addressing risk assessment in non-stationary environments, Jain et al. (2021) described a method to analyze sources of lack of knowledge by adding a second Bayesian model to predict algorithmic action risks, particularly relevant for multi-agent RL (MARL) systems. Kutschireiter et al. (2023) developed a Bayesian ring attractor that outperforms conventional ring attractors by dynamically adjusting its activity based on

evidence quality and uncertainty. In the context of individual treatment effects, Lee et al. (2020) performed uncertainty quantification (UQ) using an exogenously prescribed algorithm, making the method agnostic to the underlying recommender algorithm.

Azizzadenesheli et al. (2018) developed a Bayesian approaches for RL in episodic high-dimensional Markov decision processes (MDPs). They introduced two novel algorithms: LINUCB and LINPSRL. These algorithms achieve significant improvements in sample efficiency and performance by incorporating uncertainty estimation into the learning process. The extension to Deep RL, called Bayesian deep Q-networks, BDQNs (Azizzadenesheli et al., 2018), incorporates efficient Thompson sampling and Bayesian linear regression at the output layer to factor uncertainty estimation in the action-value estimates. On a similar line, Foerster et al. (2019) proposed a Bayesian action decoder. It is a learning algorithm based on approximate Bayesian updates to obtain a public belief that conditions the actions taken by other agents in the environment. This creates uncertainty-aware agents that are not biased by training data. It also generates a factorised, approximate belief state that provides the agents with efficient learning through informed actions.

In summary, the literature reveals a growing interest in incorporating spatial awareness, biological inspiration, and UQ into RL systems. However, there remains a gap in integrating these elements into a cohesive framework. Our work on ring attractors aims to bridge this gap by providing a biologically plausible model that inherently captures spatial relations and can be extended to handle uncertainty, potentially leading to more robust and efficient RL agents.

## 3 METHODOLOGY

In this section, we describe two main methods: an exogenous ring attractor model using continuous-time recurrent neural networks (CTRNNs) and a DL-based ring attractor integrated into the RL agent. Both leverage the ring attractors' spatial encoding capabilities to enhance action selection and performance. We detail the ring attractor architecture, dynamics, and implementation, including uncertainty injection in the CTRNN model for robust decision-making. CTRNNs are employed for their ability to model continuous neural dynamics and maintain stable attractor states (Beer, 1995). The integrated approach offers end-to-end training for efficiency and scalability. Ring attractors in RL maintain stable spatial information representations, preserving action relations lost in traditional flattened action spaces. This circular spatial representation potentially yields smoother policy gradients and more efficient learning in spatial tasks, attributed to the ring attractors' ability to maintain a stable representation of spatial information.

### 3.1 EXOGENOUS RING ATTRACTOR MODEL: CONTINUOUS-TIME RNN

During the first stage of the research, the focus is on developing a self-contained ring attractor as a CTRNN. This will be integrated into the output of the value-based policy model to perform action selection.

#### 3.1.1 RING ATTRACTOR ARCHITECTURE

Ring attractors commonly consist of a configuration of excitatory and inhibitory neurons arranged in a circular pattern. We can model the dynamics of the ring using the Touretzky ring attractor network (Touretzky, 2005). In this model, each excitatory neuron establishes connections with all other excitatory neurons, and an inhibitory neuron is placed in the middle of the ring with equal weighted connections to all excitatory neurons. This creates a network that facilitates complex information processing.

**Excitatory neurons' input signal**. Let $x_n^i \in \mathbb{R}^s$ denote the input signal from source number $i$ to the excitatory neuron $n = 1, \ldots, N$. The total input to neuron $n$ is defined as the sum of all input signals $I$ for that particular neuron: $x_n = \sum_{i=1}^{I} x_n^i$, where $x_n \in \mathbb{R}$. To model input signals $x_n^i$ of varying strengths, these signals are commonly viewed as Gaussian functions $x_n^i : \mathbb{R}^s \to \mathbb{R}^s$. These functions allow us to represent the input to each neuron as a sum of weighted Gaussian distributions. The key parameters of these Gaussian functions are: $K_i$, the magnitude variable for the input signal in index $i$, which determines the overall strength of the signal; $\mu_i$, which defines the mean position of the the Gaussian curve in the ring for the input signal $i$, representing the central focus of the signal;

$\sigma_i$, the standard deviation of the Gaussian function, which determines the spread or reliability of the signal; and $\alpha_n$, which represents the preference for the orientation of the neuron $n$ in space. These parameters combine to the following:

$$x_n(K_i) = \sum_{i=1}^{I} x_n^i(\alpha_n) = \sum_{i=1}^{I} \frac{K_i}{\sqrt{2\pi}\sigma_i} \exp\left(-\frac{1}{2}\frac{(\alpha_n - \mu_i)^2}{\sigma_i^2}\right), \quad n = 1, 2, \ldots, N \qquad (1)$$

**Neuron activation function**. We employ the rectified linear unit (ReLU) function $f(x) = \max(0, x + h)$, where $h \in \mathbb{R}^+$ as the activation function for each neuron, where $h$ is a threshold that introduces the non-linear behaviour in the ring.

**Excitatory neuron dynamics**. The dynamics of excitatory neurons in the ring is described as:

$$\frac{dv_n}{dt} \approx \frac{\Delta v_n}{\Delta t} = \frac{v_{n+\Delta t} - v_n}{\Delta t} = \frac{f(x_n + \epsilon_n + \eta_n)}{\tau} - v_n \qquad (2)$$

In Eq. 2 $v_n \in \mathbb{R}$ represents the activation of the excitatory neuron $n$, $x_n$ is previously defined in Eq. 1 is the external input to the neuron $n$ of Eq. 1, $\epsilon_n \in \mathbb{R}$ represents the weighted influence of the other excitatory neurons activation, which is defined mathematically in Eq. 4. $\eta_n \in \mathbb{R}$ is the influence from the weighted inhibitory neuron activation to the target excitatory neuron $n$, and $\tau = \Delta t$ is the time integration constant. This equation captures the evolution of neuronal activation over time, considering both excitatory and inhibitory activations.

**Inhibitory neuron dynamics**. The activation of the inhibitory neuron, which regulates network dynamics, is described by:

$$\frac{du}{dt} \approx \frac{\Delta u}{\Delta t} = \frac{u_{+\Delta t} - u}{\Delta t} = \frac{f(\epsilon_n + \eta_n)}{\tau} - u \qquad (3)$$

Here, $u \in \mathbb{R}$ represents the inhibitory neuron's activation output, $\epsilon_n$ is the weighted sum of excitatory activations where in this case $n$ is the inhibitory neuron, and $\eta_n \in \mathbb{R}$ is the weighted self-inhibition activation term. This equation models how the inhibitory neuron integrates inputs from the excitatory population and its own state.

**Synaptic weighted connections**: The influence between neurons decreases with distance, as modeled by the weighted connections. This weighted connection applies to both excitatory and inhibitory neurons. For excitatory neurons: $w^{(E_m \to E_n)} = e^{-d_{(m,n)}^2}$, where $d_{(m,n)} = |m - n|$ is the distance between neurons $m$ and $n$. For the inhibitory neuron: $w^{(I \to E_n)} = e^{-d_{(m,n)}^2} = e^{-1}$. Note that our model contains a single inhibitory neuron placed in the middle of the ring, with a distance of 1 unit to all excitatory neurons. The excitatory ($\epsilon_n$) and inhibitory ($\eta_n$) weighted connections are also known in the literature as neuron-proximal excitatory and inhibitory voltage or potential. These are then defined as follows:

$$\epsilon_n = \sum_{m=1}^{N} w_{m,n}^{(E_m \to E_n)} v_m \qquad \eta_n = w^{(I \to E_n)} u \qquad (4)$$

**Full excitatory neuron dynamics**. Combining all influences, the complete dynamics of excitatory neurons are described by:

$$\frac{dv_n}{dt} = \frac{1}{\tau}\left(f\left(\sum_{m=1}^{N} w_{m,n}^{(E_m \to E_n)} v_m + x_n + w^{(I \to E_n)} u\right)\right) - v_n \qquad (5)$$

This equation updates the activation for all neurons based on recurrent excitation, external input from the summation of the input signals for neuron $n$, $x_n$; and inhibitory influence.

**Full inhibitory neuron dynamics**. The complete dynamics of the inhibitory neuron are given by:

$$\frac{du}{dt} = \frac{1}{\tau}\left(f\left(u + \sum_{m=1}^{N} w_m^{(E_m \to I)} v_m\right)\right) - u \qquad (6)$$

This equation models how the inhibitory neuron integrates self-inhibition and excitatory inputs from the entire network. These equations collectively describe the complex dynamics of the ring attractor network, capturing the interplay between excitatory and inhibitory neurons, external inputs, and synaptic connections.

### 3.1.2 Ring Attractor as Behavior Policy in Reinforcement Learning

To integrate the ring attractor model with RL, we need to establish a connection between the estimated value of state-action pairs and the input to the ring attractor network. This integration allows the ring attractor to serve as a behavior policy, guiding action selection based on the values learned. We begin by reformulating the input function for a target excitatory neuron $n$. The key modification is setting the scale factor $K_i$ to the Q-value $Q(s, a)$ of the state-action pair $(s, a)$, that is $K_i = Q(s, a)$.

This formulation ensures that actions with higher estimated values are given more weight in the ring attractor dynamics, naturally biasing the network towards more valuable actions. The orientation of the signal within the ring attractor is determined by the direction of movement in the action space. We represent this as $\mu_i = \alpha_a(a)$, where $\alpha_a(a)$ is the angle corresponding to the action $a$ in the circular action space. We define our circular action space $\mathcal{A}$ as a subset of $\mathbb{R}^2$, where each action $a \in \mathcal{A}$ is represented by a point on the unit circle. The function $\alpha : \mathcal{A} \rightarrow [0, 2\pi]$ maps each action to its corresponding angle on this circle, and $\alpha_n$ which presents the preference for the orientation of the neuron $n$ in space. To account for uncertainty in our value estimates, we incorporate the variance of the estimated value for each action into our model: $\sigma_i = \sigma_a$.

This allows the network to represent not just the expected value of actions, but also our confidence in those estimates. Combining these elements, we arrive at the following equation for the action signal $x(a)$:

$$x_n(Q) = \sum_{a=1}^{A} \frac{Q(s, a)}{\sqrt{2\pi\sigma_a}} \exp\left(-\frac{1}{2}\frac{(\alpha_n - \alpha_a(a))^2}{\sigma_a^2}\right) \tag{7}$$

This equation represents the input to each neuron as a sum of Gaussian functions, where each function is centered on an action's direction and scaled by its Q-value. The dynamics of the excitatory neurons in the ring attractor, now incorporating the Q-value inputs, are described by:

$$\frac{dv_n}{dt} = \frac{1}{\tau}\left(max\left(0, \left(\sum_{m=1}^{m=N} w_{m,n}^{(E \rightarrow E)}v_m + x_n(Q) + w^{(I \rightarrow E_u)}u\right)\right)\right) - v_n \tag{8}$$

This equation captures how the activation of each neuron evolves over time, influenced by the action-value functions $x_n(Q)$, and both excitatory and inhibitory feedback. This equation captures how the activation of each neuron evolves over time, influenced by the action-value input Gaussian functions $x_n(Q)$, the excitatory feedback $e_n = \sum_{m=1}^{N} w^{(E_m \rightarrow E_n)}v_m$, and the inhibitory feedback $i_n = w^{(I \rightarrow E_n)}u$.

To translate the ring attractor's output into an action in the 2D space, we use the following equation:

$$action = \arg\max_{n}\{\mathbf{V}\} \cdot \frac{N^{(A)}}{N^{(E)}} \tag{9}$$

where $n \in \{1, ..., N^{(E)}\}$, $N^{(E)}$ is the number of excitatory neurons in the ring attractor, $N^{(A)}$ is the number of discrete actions in the action space $\mathcal{A}$, $\mathbf{V} = [v_1, v_2, ..., v_{N^{(E)}}]$.

This equation assumes that both the neurons in the ring attractor and the actions in the action space are uniformly distributed. This approach allows for nuanced action selection that takes into account both the spatial relations between actions and their estimated values. A visualization of the ring is presented in Fig. 1.

### 3.1.3 Uncertainty Quantification Model

In the field of DRL, for any state-action pair $(s, a)$, the Q-value $Q(s, a)$ can be expressed as a function of the input state through a function approximation algorithm $\Phi_\theta(s)$ taking as input the current state $s$. This function approximation algorithm ($\Phi_\theta(s)$) can be expressed as the weight matrix of our function approximation algorithm transposed $\theta^T$ times the feature vector extracted from the input state $x(s)$: $Q(s, a) = \Phi_\theta(s) = \theta^T x(s)$ (Sutton & Barto, 2018). As stated in Section 3.1.2, the variance of the Gaussian functions, input to the ring attractor, will be given by the variance of the estimate value for that particular action $\sigma_i = \sigma_a$.

Among the diverse methods to compute the uncertainty of the action ($\sigma_a$) we have chosen to compute a posterior distribution with Bayesian linear regression (BLR). BLR acts as output layer for our neural network (NN) of choice. We choose a linear regression model because it does not compromise the efficiency of the NN, while at the same time it provides a distribution to compute the variance for the state-action pairs. The implementation is based on Azizzadenesheli et al. (2018), where a Bayesian value-based DQN model was instantiated to output an uncertainty-aware prediction for the state-action pairs. With this approach, the new $Q$ function is defined as:

$$Q(s, a) = \Phi_\theta(s)^T w_a, \tag{10}$$

where $w_a$ are the weights from the posterior distribution of the BLR model. Eq. 10 represents the parameters of the final Bayesian linear layer.

When provided with a state transition tuple $(s, a, r, s')$, where $s$ is the current state, $a$ is the action taken, $r$ is the reward received, and $s'$ is the next state. This tuple represents a single step of interaction between the agent and the

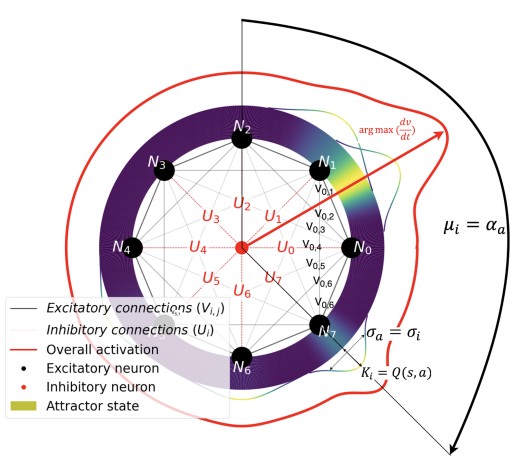

Figure 1: Ring attractor Touretzky representation: Circular arrangement of excitatory neurons (N0-N7) with excitatory connections and central inhibitory neuron. Four input signals shown as colored gradients. Overall activation depicted by red outline. Includes connection weights and input signal parameters, illustrating ring attractor dynamics.

environment in the RL framework. The model learns to adjust the weights $w_a$ of the BLR and the function approximation algorithm, i.e. neural networks (NNs) ($\Phi_\theta$), to align the Q values with the optimal action $a = argmax(\gamma \Phi_\theta(s)^T w_a)$, equation 11.

$$Q(s, a) = \Phi_\theta(s)^T w_a \rightarrow y := r + \gamma \Phi_{\theta_{\text{target}}}(s')^T w_{\text{target}} \hat{a} \tag{11}$$

where $\gamma$ is the discount factor, $y$ is the expected Q-value, $\Phi_{\theta_{\text{target}}}$ are the features from the next state $s'$ extracted by the function approximation algorithm $\Phi$ using the target network parameters, $\theta_{target}$ refers to the parameters of the target function approximation algorithm used for learning, and $\hat{a}$ is the predicted optimal action in the next state $s'$. The construction of the Gaussian BLR prior distribution and the weights $i$ sample $w_{a,i}$ collected from the posterior distribution are performed through Thompson Sampling. This process allows us to incorporate uncertainty into our action-value estimates. For details on the construction of the Gaussian prior distribution and the specifics of the sampling process, we refer readers to Azizzadenesheli et al. (2018). Both the mean $\bar{Q}(s, a)$ and the variance $\bar{\sigma}_a^2$ from Eq. 12 are calculated from a finite number of samples $I$.

$$\bar{Q}(s, a) = \frac{\sum_{i=0}^{i=I} Q(s, a)_i}{I} = \frac{\sum_{i=0}^{i=I} w_{a,i}^T \Phi_\theta(s_t)}{I}$$

$$\bar{\sigma}_a^2 = \frac{\sum_{i=0}^{i=I} \left( w_{a,i}^T \Phi_\theta(s_t) - \mu_a \right)^2}{I - 1} \tag{12}$$

## 3.2 DEEP LEARNING RING ATTRACTOR MODEL

To further enhance the ring attractor's integration into RL frameworks and agents, we provide a Deep Learning (DL) implementation. This approach improves model learning and integration with DRL agents. Our implementation offers both algorithmic improvements, by benefiting from DL training process, and software integration improvements, easing the deployment processes. Recurrent Neural Networks (RNNs) offer a practical approach for integrating ring attractors within DRL agents. Recent studies by Li et al. (2015) show that RNNs perform well in modeling sequential data and in capturing temporal dependencies for decision-making. Like CTRNNs, RNNs mirror ring attractors' temporal dynamics, with their recurrent connections and flexible architecture emulating the interconnected nature of ring attractor neurons. This allows modeling of weighted connections for

both forward and recurrent hidden states, as shown in the Appendix A.3. The premises for modeling the structure of the RNN are as follows.

**Attractor state as recurrent connections**. RNN recurrent connections model the attractor state, integrating information from previous time steps into the current network state, allowing retention of information over time.

**Signal input as a forward pass**. Forward connections from previous layers are arranged circularly, mimicking the ring's spatial distribution. The attractor state encodes task context, influenced by current input and hidden state. A learnable time constant $\tau$, inherited from Eq. 2, controls input contributions and temporal evolution, enabling adaptive behavior and adaptable input contribution to the attractor state.

### 3.2.1 DEEP REINFORCEMENT LEARNING AGENT INTEGRATION

To shape circular connectivity within a RNN, the weighted connections in the input signal $V(s)$ and the hidden state or attractor state $U(v)$ are computed as follows:

$$
\begin{aligned}
V(s)_{m,n} &= \frac{1}{\tau} \Phi_\theta(s)^T w_{m,n}^{I \to H} = \frac{1}{\tau} \Phi_\theta(s)^T e^{\frac{d(m,n)}{\lambda}} \\
d(m,n) &= \min\left(|m - n\frac{M}{N}|, N - |m - n\frac{M}{N}|\right) \\
U(v)_{m,n} &= h(v)^T w_{m,n}^{H \to H} = h(\Phi_\theta)^T e^{\frac{d(m,n)}{\lambda}} \\
d(m,n) &= \min\left(|m - n|, N - |m - n|\right)
\end{aligned}
\tag{13}
$$

This circular structure mimics the arrangement of excitatory neurons in the ring attractor. Eq. 13 shows the input signal to the recurrent layer $V_{m,n}$ from neuron $m$ from the previous layer in the DL agent to neuron $n$ in the RNN. The hidden state, $U_{m,n}$ mimics an attractor state, representing the recurrent connections in the RNN. The weighted RNN connections include fixed input-to-hidden connections ($w^{I \to H}m, n$) to maintain the ring's spatial structure, and learnable hidden-to-hidden connections ($w^{H \to H}m, n$) to capture emerging action relationships. These depend on a parameter $\lambda$ that drives the decay of the potential over distance and distance between neurons $d(m,n)$ where $N$ is the total number of neurons for the RNN and $M$ is the count of neurons in the previous layer of the NN architecture. The function $\phi_\theta(s) : \mathbb{R}^S \to \mathbb{R}^M$ maps the input state $s$ of the DL agent to a representation of characteristics that will be the input of the recurrent layer. Likewise, $\theta$ represents the parameters of this function (i.e., the weights and biases of the NN layers preceding the RNN layer, which extract relevant features from the input). The function $h(v) : \mathbb{R}^N \to \mathbb{R}^N$ is a parameterized by learnable weights transformation that maps the information from previous forward passes into the current hidden state. The learnable parameter $\tau$ is the positive time constant responsible for the integration of signals in the ring. It defines the contribution of input states $\phi_\theta(s)$ to the current hidden state, imitating the attractor state, applied to neural networks.

Finally, the action-value function $Q(s,a)$ is derived from the RNN layer's output by applying the neurons activation function to the combined input $V(s)$ and hidden state information $U(s)$. The activation function of choice is a hyperbolic tangent $\tanh$, this function is symmetric around zero, leading to faster convergence and stability. However, the output range of $\tanh$ (-1 to 1) is not fully compatible with value-based methods, where the DL agents needs to output action-value pairs in the range of the environment's reward function. To address this issue and prevent saturation of the $\tanh$ activation function, we scale the action-value pairs by multiplying them with a learnable scalar $\beta$, as $Q(s,a) = \beta h_t = \beta \tanh((V(s) + U(v)) = \beta \tanh((\frac{1}{\tau} \Phi_\theta(s_t)^T w^{I \to H} + h_{t-1}(v)^T w^{H \to H}))$.

## 4 EXPERIMENTS

This section presents the findings from our experiments that validate our proposed approach of integrating ring attractors into RL algorithms. To assess the effectiveness of our method, we conduct comparisons across multiple baseline models and action spaces. The evaluation encompasses two implementations: a traditional exogenous ring attractor and an innovative approach where the ring attractor is modeled directly into a DRL agents. In both implementations, action-value pairs $Q(s,a)$ are evenly distributed across the ring circumference. For the exogenous model, each action is associated with a specific angle on the ring, Section 3.1.2. In the DL implementation, each neuron in

the RNN corresponds to one action-value. The ring attractor serves as the output layer of the DL agent with the weights modeling the circular topology of the action space, Section 3.2.1. For both approaches, agents are annotated with the suffix $RA$.

Our results demonstrate the effectiveness of ring attractors in enhancing action selection and significantly speeding up the learning process of the Reinforcement Learning agent overall.

## 4.1 Exogenous Ring Attractor Model Performance Analysis

To evaluate our exogenous ring attractor model integrated with BDQN (Azizzadenesheli et al., 2018) we performed experiments in the OpenAI Super Mario Bros environment (Kauten, 2018). This benchmark exhibits an spatially distributed complex decision-making scenario.

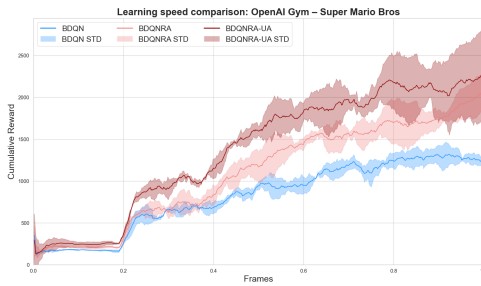 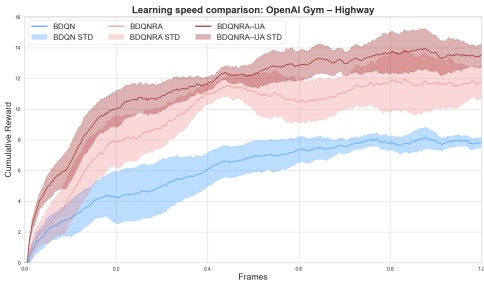

Figure 2: Learning speed comparison: Right OpenAI Gym Super Mario Bros environment (Kauten, 2018) with discrete action space; Left: OpenAI highway (Leurent, 2018), with a continuous 1-D circular variable. The plot shows cumulative reward over 1 million frames for three models: Standard BDQN; BDQNRA with ring attractor behavior policy from Section 3.1.2, setting the action-value pair variance constant to $\sigma_a = \frac{\pi}{6}$, using this fix variance to enable smooth action transitions while preventing interference with opposing actions; and BDQNRA-UA with RA and Uncertainty Awareness (UA) implementing the uncertainty quantification model from 3.1.3 to feed into the variance of the action-value pairs. Displaying mean episodic returns over 10 averaged seeds.

Fig. 2 shows that both ring attractor models (BDQNRA and BDQNRA-UA) consistently outperform standard BDQN. The uncertainty-aware version (BDQNRA-UA) shows the best overall performance, highlighting the benefits of combining ring attractors spatial distribution of the action space with uncertainty-aware action selection. Empirical evaluations revealed that the CTRNN-based ring attractor models exhibited a mean computational overhead of 297.3% (SD = 14.2%) compared to the baseline, significantly impacting runtime efficiency. To address this performance bottleneck and integrating the ring attractor spatial understanding into the DRL, we developed a DL implementation of the ring attractor. This DL implementation is evaluated in the subsections below.

## 4.2 Deep Learning Ring Attractor Model Performance Analysis

This subsection details the effects of incorporating uncertainty quantification through Bayesian Linear Regression. We evaluate the quality of uncertainty estimates and their impact on exploration strategies and overall agent performance. Fig. 3, shows that the DDQNRA model consistently outperforms the standard DDQN across tasks for both navigation and game-like decision-making scenarios. This suggests that the ring attractor's ability to encode spatial relationships in different actions spaces contributes significantly to the agent's learning efficiency. These results indicate that the integration of ring attractors into DRL architectures can lead to significant improvements in both learning speed and overall performance, especially in environments with strong spatial components.

## 4.3 Performance on Atari 100k Benchmark

In this results section, we provide a comprehensive analysis of our model's performance on the Atari 100k benchmark (Bellemare et al., 2012). We present detailed comparisons with state-of-the-art models, highlighting the improvements achieved by our approach. We analyze the factors

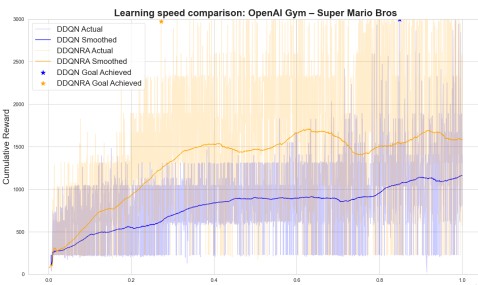 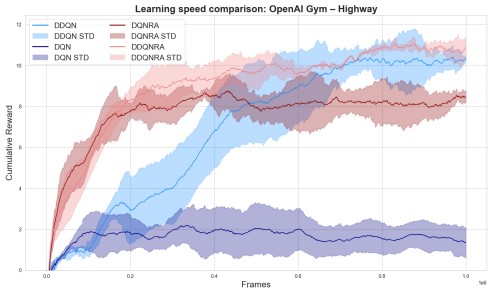

Figure 3: Performance comparison: DDQNRA vs standard DDQN (van Hasselt et al., 2015) in two environments. Right: OpenAI highway (Leurent, 2018), showing learning speed in spatial navigation tasks. Left: OpenAI Super Mario Bros (Kauten, 2018), demonstrating adaptability to complex, game-like scenarios. Displaying mean episodic returns over 10 averaged seeds.

contributing to the significant performance increase mentioned in the introduction, breaking down results by game, and discussing notable trends or patterns observed across different types of tasks.

Table 1: Performance comparison on Atari 100k Benchmark (Bellemare et al., 2012). Benchmark performed across all environments where actions can be layout in one or more 2D action space planes (ring attractors). This is represented by the *ring* configuration column. The results are recorded at the end of training and averaged over 10 random seeds, 3 samples per seed. We display game score and overall mean and median human-normalized scores for each algorithm.

| Game | | Agent: | Reported | | | Implemented | |
|---|---|---|---|---|---|---|---|
| Environment | Ring | Human | CURL | SPR | EffZero | EffZero | EffZeroRA |
| Alien | Double | 7127.7 | 558.2 | 801.5 | 808.5 | 738.1 | **1098.8** |
| Asterix | Single | 8503.3 | 734.5 | 977.8 | 25557.8 | 14839.3 | **31037.3** |
| Bank Heist | Double | 753.1 | 131.6 | 380.9 | 351.0 | 362.8 | **460.5** |
| BattleZone | Double | 37187.5 | 14870.0 | **16651.0** | 13871.2 | 11908.7 | 15672.0 |
| Boxing | Double | 12.1 | 1.2 | 35.8 | 52.7 | 30.5 | **62.4** |
| Chopper C. | Double | 7387.8 | 1058.5 | 974.8 | 1117.3 | 1162.4 | **1963.0** |
| Crazy Climber | Single | 35829.4 | 12146.5 | 42923.6 | 83940.2 | 83883.0 | **100649.7** |
| Freeway | Double | 29.6 | 26.7 | 24.4 | 21.8 | 22.7 | **31.3** |
| Frostbite | Double | 4334.7 | 1181.3 | 1821.5 | 296.3 | 287.5 | **354.8** |
| Gopher | Double | 2412.5 | 669.3 | 715.2 | 3260.3 | 2975.3 | **3804.0** |
| Hero | Double | 30826.4 | 6279.3 | 7019.2 | 9315.9 | 9966.4 | **11976.1** |
| Jamesbond | Double | 302.8 | 471.0 | 365.4 | **517.0** | 350.1 | 416.4 |
| Kangaroo | Double | 3035.0 | 872.5 | 3276.4 | 724.1 | 689.2 | **1368.8** |
| Krull | Double | 2665.5 | 4229.6 | 3688.9 | 5663.3 | 6128.3 | **9282.1** |
| Kung Fu M. | Double | 22736.3 | 14307.8 | 13192.7 | 30944.8 | 27445.6 | **49697.7** |
| Ms Pacman | Single | 6951.6 | 1465.5 | 1313.2 | 1281.2 | 1166.2 | **2028.0** |
| Private Eye | Double | 69571.3 | **218.4** | 124.0 | 96.7 | 94.3 | 155.8 |
| Road Runner | Double | 7845.0 | 5661.0 | 669.1 | 17751.3 | 19203.1 | **29389.3** |
| Seaquest | Double | 42054.7 | 384.5 | 583.1 | 1100.2 | 1154.7 | **1532.8** |
| **Human–normalised Score** | | | | | | | |
| | Mean | 1.000 | 0.428 | 0.638 | 1.101 | 0.959 | **1.454** |
| | Median | 1.000 | 0.242 | 0.434 | 0.420 | 0.403 | **0.531** |

Table 1 presents a comprehensive comparison of our ring attractor-based RL model integrated with Efficient Zero (Ye et al., 2021), evaluating performance across multiple Atari games with a limited training budget of 100,000 environment steps. The table includes results from baseline methods and recent top-performing algorithms SPR (Schwarzer et al., 2020) and CURL (Srinivas et al., 2020) for context. Our model demonstrates significant improvements over baseline methods, particularly

in games with inherent spatial components, such as Asterix and Boxing, showing 110% and 105% improvement respectively over the previous state-of-the-art.

The mapping between game action spaces and ring configurations reflects the fundamental structure of each environment's action space. Games with primarily directional movement actions, such as *Asterix* and *Ms Pacman*, utilize a *Single* ring configuration where eight directional movements map naturally to positions around the ring circumference. In contrast, games combining movement with independent action dimensions, such as *Seaquest* and *BattleZone*, employ a *Double* ring configuration, one ring encoding movement actions and another representing secondary mechanics such as combat. This architecture maintains spatial relationships while preserving the independence of different action types. Further implementation details for multiple ring dynamics can be found in Appendix A.4.

These results reaffirm that the spatial encoding provided by ring attractors is especially beneficial in environments where spatial relationships between actions are key. The consistent performance improvement different games indicates that our approach provides a general enhancement that benefits a wide range of RL tasks. Even in games where the improvement is less dramatic, we still see substantial increases in performance, suggesting that the benefits of the ring attractor extend beyond just spatially-oriented games.

To ensure a fair comparison under identical experimental conditions, we re-implemented and evaluated both the baseline EffZero and our proposed EffZeroRA model using the same computational resources and experimental setup as employed throughout this study.

Ablation studies were conducted to isolate the impact of key components in our ring attractor models, detailed in Appendix A.2.1 and A.2.2. For the exogenous model, we compared performance with correct and randomized action distributions in the ring. In the DL implementation, we removed the circular weight distribution to assess its importance.

## 5 CONCLUSION

This paper presents a novel approach to RL, integrating ring attractors into action selection. Our work demonstrates that these neuroscience-inspired ring attractors significantly enhance learning capabilities for value-based RL agents, leading to more stable and efficient action selection, particularly in spatially structured tasks.

### 5.1 KEY FINDINGS AND IMPLICATIONS

The integration of ring attractors as a DL module proves particularly effective, allowing for end-to-end training and easy incorporation into existing RL architectures. This approach improves performance and offers potential insight into explicit spatial encoding of actions.

Our results demonstrate significant improvements in action selection and learning speed. We achieve state-of-the-art performance on the Atari 100k benchmark, with an average 53% performance increase across all games tested compared to the baseline and previous state-of-the-art models. Notable improvements were observed in games with strong spatial components, such as Asterix (110 % improvement) and Boxing (105% improvement). Additionally, we observed improvements in other environments tested outside the Atari benchmark, further supporting the effectiveness of our approach across various RL tasks and agents.

### 5.2 FUTURE WORK

We acknowledge that our approach, while promising, has limitations and areas for potential improvement. Future research should investigate the scalability of this method in high-dimensional action spaces and explore its efficacy in domains where spatial relationships are less straightforward.

We believe that the success of this approach opens up several future research paths. The current work can be extended to multi-agent scenarios and policy-based RL agents. In the field of uncertainty-aware decision making, leveraging the spatial structure provided by attractor networks presents a promising avenue to map uncertainty explicitly to the action space. Deploying the techniques pre-

sented here into specific domains could yield performance boosts, especially in safe RL, leveraging their stability properties to enforce constraints and ensure predictable behavior.

This approach not only improves performance but also offers potential insight into spatial encoding of actions and decision-making processes, bridging the gap between neuroscience-inspired models and practical RL agents.

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

# A  APPENDIX

## A.1  BACKGROUND:ATTRACTOR NETWORKS THEORETICAL FOUNDATIONS

Ring attractor networks are a type of biological neural structure that has been proposed to underlie the representation of various cognitive functions, including spatial navigation, working memory, and decision-making (Kim et al., 2017).

**Biological intuition**. In the early 1990s, the research carried out by Zhang (1996) proposed that ring neural structures could underlie the representation of heading direction in rodents. Zhang (1996) argued that the neural activity in an attractor network might encode the direction of the animal's head, with the network transitioning from one attractor state to another state as the animal turns.

**Empirical evidence**. There is growing evidence from neuroscience supporting the role of ring attractors in neural processing. For example, electrophysiological recordings from head direction cells (HDCs) of rodents have revealed a circular organisation of these neurons, with neighbouring HDCs encoding slightly different heading directions (Taube, 1995). Furthermore, studies have shown that HDC activity can be influenced by sensory inputs, such as visual signals and vestibular signals, and that these inputs can cause the network to update its representation of heading direction (Taube, 2007). Xiong & Sridhar (2024) showed model internal noise biases toward accuracy over speed, while environmental uncertainty exhibits a U-shaped effect where moderate uncertainty favors speed and extreme uncertainty favors accuracy. Wilson (2023) demonstrated that biological navigation networks rely on attractor dynamics while maintaining adaptability through continuous synaptic plasticity and sensory feedback.

**Sensor fusion in ring attractors**. Ring attractor networks provide a theoretical foundation for understanding cognitive functions such as spatial navigation, working memory, and decision-making. In the context of action selection in RL, sensor fusion plays a pivotal role in augmenting the information-processing capabilities of these networks. By combining data from various sensory modalities, ring attractors create a more nuanced and robust representation of the environment, essential for adaptive behaviors (ME, 2008). Research has elucidated the relationship between ring attractors and sensory inputs, with the circular organisation of HDCs in rodents complemented by the convergence of visual and vestibular inputs, highlighting the integrative nature of sensory information within the ring attractor framework (Zugaro et al., 2001).

**Modulation by sensory inputs**. Beyond the spatial domain, sensory input dynamically influences the activity of ring attractor networks. Studies have shown that visual cues and vestibular signals not only update the representation of heading direction but also contribute to the stability of attractor states, allowing robust spatial memory and navigation (Goodridge et al., 1998).

**Sensor fusion for action selection**. The concept of sensor fusion within the context of ring attractors extends beyond traditional sensory modalities, encompassing diverse sources, such as proprioceptive and contextual cues (McNaughton et al., 1996). Building on the foundation of ring attractor networks discussed earlier, the integration of sensor fusion in the context of action selection involves fusing the action values associated with each potential action within the ring attractor framework. In particular, sensory information, previously shown to modulate the activity of ring attractor networks, extends its influence to the representation of action values. The inclusion of sensory information reflects a higher cognitive process, where the adaptable nature of ring attractor networks plays a central role in orchestrating optimal decision making and action selection in complex environments.

## A.2  VALIDATING RING ATTRACTOR CONTRIBUTIONS THROUGH ABLATION STUDIES

### A.2.1  EXOGENOUS RING ATTRACTOR MODEL ABLATION STUDY

To isolate the impact of the ring attractor structure, we conducted an ablation study comparing our full BDQNRA model against versions with the action space overlay in an incorrect distribution in the ring, Fig. 4. This incorrect distribution involves randomly rearranging the placement of actions within the ring, disrupting the natural topology of the action space.

For instance, this could mean placing opposing or unrelated actions side by side in the ring, such as pairing "move left" with "move down" instead of its natural opposite "move right". More gen-

erally, this incorrect distribution breaks the inherent relationships between actions that are typically preserved in the ring structure.

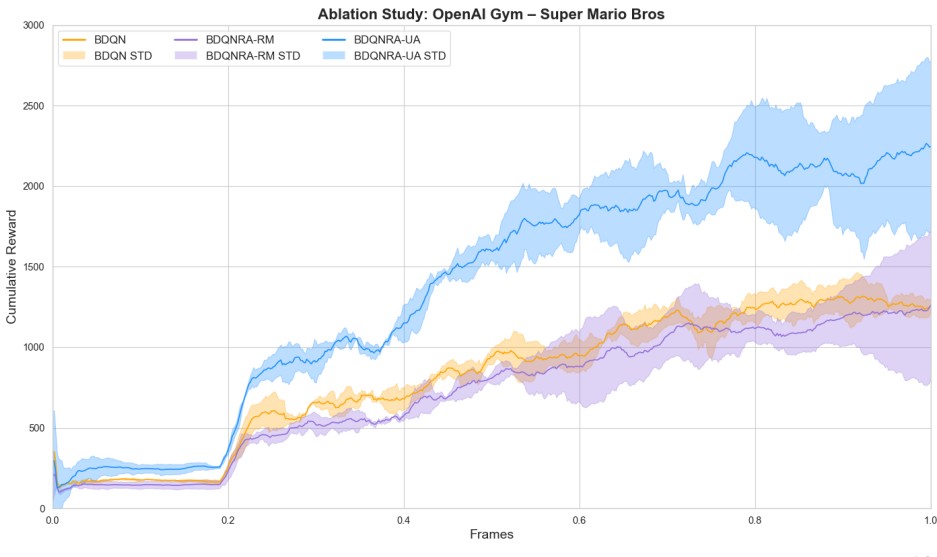

Figure 4: Ablation study comparing BDQN variants in OpenAI Gym Super Mario Bros (Kauten, 2018). The plot shows cummulative reward over 1 million frames for three models: Standard BDQN (Azizzadenesheli et al., 2018) ; BDQNRA-UA with RA and Uncertainty Awareness (UA) implementing both the ring attractor behavior policy from Section 3.1.2 and the uncertainty quantification model from 3.1.3; and BDQNRA-RM, applying the same concepts from BDQNRA-UA, but randomly distributing the action space across the ring in each experiment. Displaying mean episodic returns over 10 averaged seeds.

### A.2.2 DEEP LEARNING RING ATTRACTOR MODEL ABLATION STUDY

This ablation study focused on isolating the impact of the ring-shaped connectivity in our RNN-based ring attractor model. The key aspect of our experiment was to remove the circular weight distribution in both the forward pass (input-to-hidden connections) and the recurrent connections (hidden-to-hidden), while maintaining all other aspects of the RNN architecture. This approach allows us to directly assess the contribution of the spatial ring structure to the model's performance.

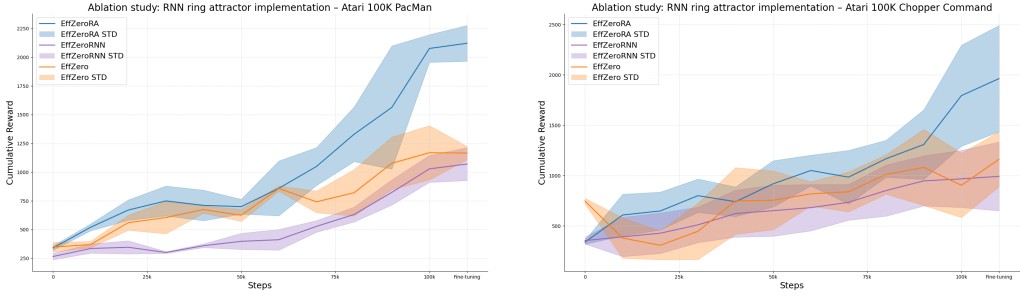

Figure 5: Ablation study results comparing the performance of the full RNN-based ring attractor model against a version with the circular weight distribution removed. The graph illustrates a significant performance drop for the Ms Pacman and Chopper Command environments in the Atari 100K benchmark (Bellemare et al., 2012). This emphasises the role of the circular topology in encoding spatial information and enhancing learning. Displaying mean episodic returns over 10 averaged seeds.

In our original model, the weights between neurons were determined by a distance-dependent function that created a circular topology. This function assigned stronger connections between neurons that were close together in the ring and weaker connections between distant neurons. For the ablation, we replaced this distance-dependent weight function with standard weight matrices for both the input-to-hidden and hidden-to-hidden connections. This modification effectively transforms our ring attractor RNN into a standard RNN, where the weights are not constrained by the circular topology. We retained other key elements of the model, such as the learnable time constant and the non-linear transformation, to isolate the effect of the ring structure specifically.

### A.2.3 DEEP LEARNING RING ATTRACTOR MODEL EVOLUTION

In this appendix section, we analyse model dynamics with both forward pass ($V(s)$) and hidden-to-hidden ($U(v)$) weights made trainable, rather than the standard approach of fixed forward pass connections, as presented in Section 3.2. As shown in Fig. 6, the forward pass connections preserve the ring structure over training time, with strong distance-dependent decay patterns maintained throughout the learning process. This may indicate that the network naturally favors maintaining spatial topology for transmitting sensory information on a per-frame basis.

The hidden-to-hidden connections, depicted in Fig. 7, demonstrate markedly different behavior. These connections evolve beyond their initial ring structure, developing specialized patterns that enable the encoding of environment-specific relationships between neurons in the hidden space. This flexibility in hidden layer connectivity supports the learning of complex action relationships while building upon the structured spatial representation from the forward pass.

These findings validate our standard implementation approach described in Section 3.2, where forward pass connections are fixed and only hidden-to-hidden weights remain trainable. The natural preservation of ring structure in trainable forward weights may suggest this topology is inherently beneficial for processing spatial information, while adaptable hidden weights enable the task-specific learning demonstrated in our experimental results, Section 4.2.

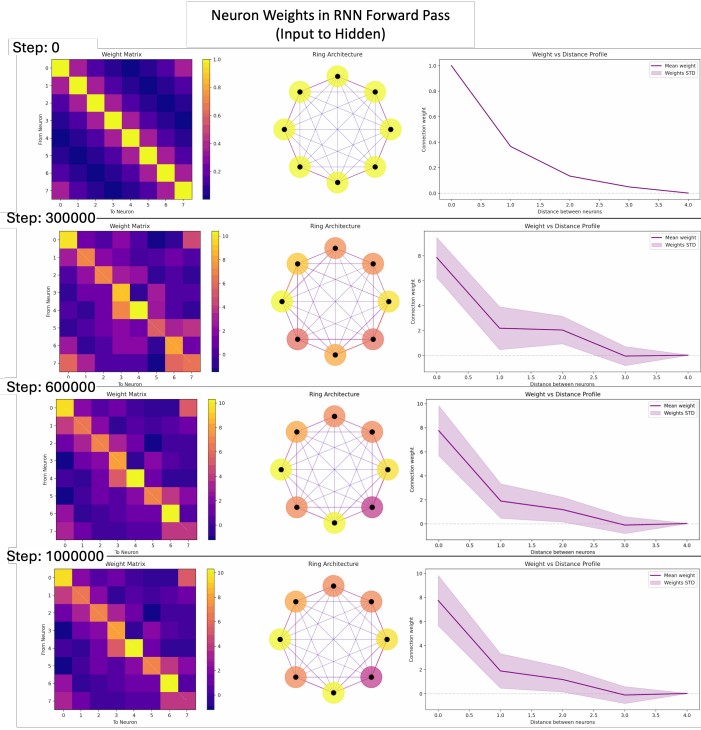

Figure 6: Evolution of forward pass weights showing preserved distance-dependent decay over training time, maintaining ring structure for spatial information transmission.

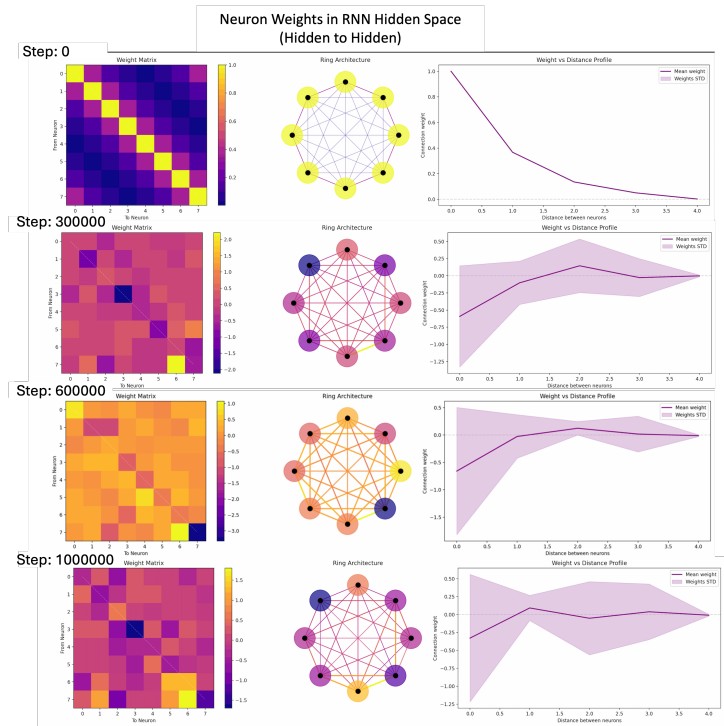

Figure 7: Development of hidden-to-hidden connections over time, demonstrating emergence of learned relationships between neurons beyond initial ring topology.

918
919
920
921
922
923
924
925
926
927
928
929
930
931
932
933
934
935
936
937
938
939
940
941
942
943
944
945
946
947
948
949
950
951
952
953

### A.3  DEEP LEARNING RING ATTRACTOR RECURRENT NEURAL NETWORK MODELING

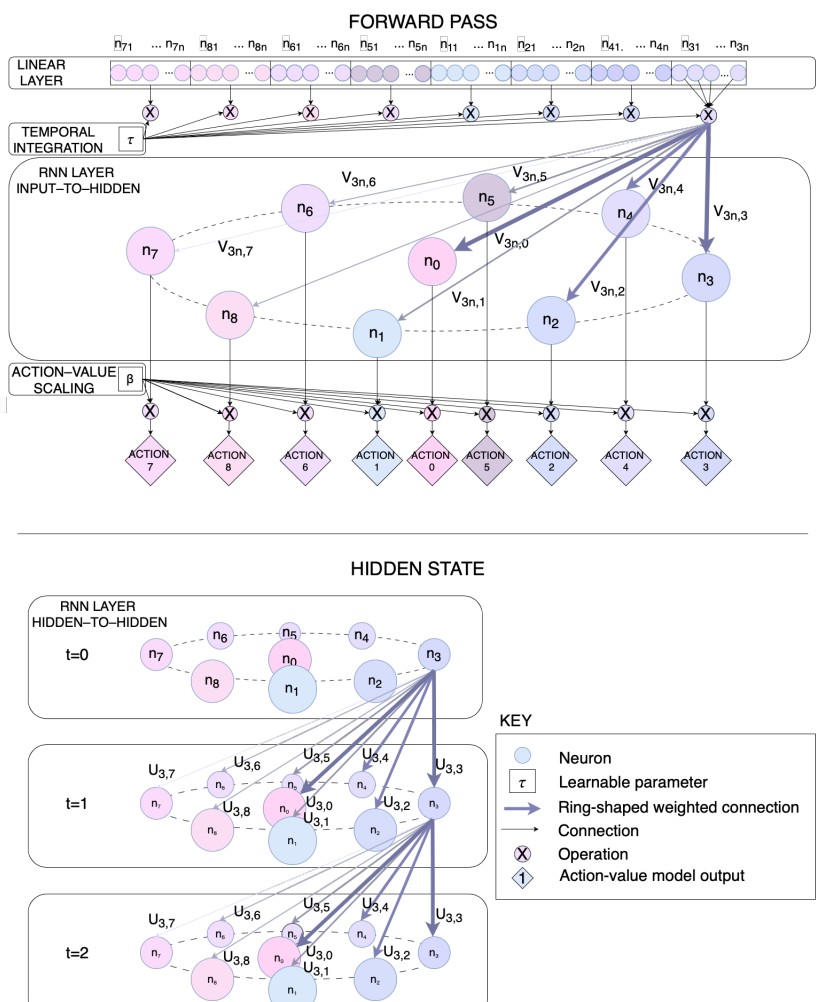

Figure 8: RNN modeling ring attractor synaptic connections: The left shows the forward pass (input-to-hidden) as the agent output layer. The right depicts the hidden-to-hidden recurrent connection between inference time steps. The weighted connections of the sample for the neuron $n3$ demonstrated.

954
955
956
957
958
959
960
961
962
963
964

As seen before, excitatory neurons are organized in a circular pattern, with connection weights between neurons determined by a distance-weighted function mimicking the synaptic connection of biological neurons, as shown in Fig.8 is the structured connectivity of the RNN, which mimics the circular topology of biological ring attractors.

965
966

### A.4  DEEP LEARNING RING ATTRACTOR MODEL IMPLEMENTATION DETAILS

967
968
969
970
971

The implementation of the ring attractor follows the equations presented in Section 3.2, where both the input-to-hidden connections $V(s)$ and hidden-to-hidden connections $U(v)$ are constructed using the distance-dependent weight functions defined in Eq. 13. These equations establish the circular topology of the ring attractor and determine how information flows through the network. However, a special case arises when dealing with neutral actions in certain Atari games, requiring a modification to the standard distance function.

### A.4.1 NEUTRAL INHIBITORY ACTION IMPLEMENTATION

For games in the Atari benchmark with neutral actions (like 'no-op'), the ring attractor maintains its circular structure with a neutral action positioned centrally. This central position creates equal connections of strength 1 to all other actions in the ring, as if it were a direct neighbor to each action simultaneously. The distance between the neutral action $n$ and any other action $m$ is fixed at 1:

$$d(m,n) = \begin{cases} 1, & \text{if } m \text{ or } n \text{ is the neutral action} \\ d(m,n), & \text{otherwise} \end{cases} \tag{14}$$

where $d(m,n)$ remains as defined in Eq. 13 for all other action pairs. The weight matrices $w_{I\to H}$ and $w_{H\to H}$ maintain the same exponential decay based on this distance function.

For example, in games like Seaquest or Asterix, this central positioning means the 'no-op' action has consistent, strong connections to all directional actions. This arrangement preserves the spatial relationships between directional actions while ensuring the neutral action remains equally accessible from any game state. The constant distance of 1 to all other actions makes transitioning to or from the neutral action as natural as moving between adjacent directional actions in the ring.

### A.4.2 DEEP LEARNING DOUBLE RING ATTRACTOR EQUATIONS

For a double ring configuration in our DL implementation as presented in thee experiments, Section 4.2, the weighted connections are defined as follows:

Input Signal to Hidden Layer (Forward Pass)

Let $\mathbf{V}^{double} \in \mathbb{R}^{2N \times 2M}$ be the complete input-to-hidden weight matrix for both rings, where $N$ is the number of output neurons per ring and $M$ is the number of input features per ring. The matrix is structured as:

$$\mathbf{V}^{double} = \begin{bmatrix} \mathbf{V}_{11} & \kappa\mathbf{V}_{12} \\ \kappa\mathbf{V}_{21} & \mathbf{V}_{22} \end{bmatrix} \tag{15}$$

where $\mathbf{V}11 = \mathbf{V}22 = \mathbf{V}12 = \mathbf{V}21$, $\kappa = 0.1$ is the cross-coupling learnable parameter initialised to 0.1. This allows the network to learn the optimal strength of interaction between the two rings during training.

Developing from Eq. 13, each ring maintains identical connectivity patterns, preserving the spatial relationships of their respective action dimensions, each submatrix $\mathbf{V}_{ij}$ represents:

$$[\mathbf{V}_{ii}]_{m,n} = \frac{1}{\tau}\Phi_\theta(s)^T e^{d(m,n)/\lambda} \tag{16}$$

where $d(m,n)$ is defined as per the forward pass weighted connections in Eq. 13.

Similarly, let $\mathbf{U}^{double} \in \mathbb{R}^{2N \times 2N}$ be the complete hidden-to-hidden weight matrix:

$$\mathbf{U}^{double} = \begin{bmatrix} \mathbf{U}_{11} & \kappa\mathbf{U}_{12} \\ \kappa\mathbf{U}_{21} & \mathbf{U}_{22} \end{bmatrix} \tag{17}$$

For primary connections ($\mathbf{U}_{11}$ and $\mathbf{U}_{22}$):

$$[\mathbf{U}_{ii}]_{m,n} = h(v)^T e^{d(m,n)/\lambda} \tag{18}$$

where $d(m,n)$ is defined as per the hidden state weighted connections in Eq. 13.

The complete forward pass for both rings is given by:

$$Q(s,a) = \beta \tanh\left(\begin{bmatrix} \frac{1}{\tau}\Phi_\theta(s_t)^T\mathbf{V}_{11} + h_{t-1}(v)^T\mathbf{U}_{11} & \frac{\kappa}{\tau}\Phi_\theta(s_t)^T\mathbf{V}_{12} + \kappa h_{t-1}(v)^T\mathbf{U}_{12} \\ \frac{\kappa}{\tau}\Phi_\theta(s_t)^T\mathbf{V}_{21} + \kappa h_{t-1}(v)^T\mathbf{U}_{21} & \frac{1}{\tau}\Phi_\theta(s_t)^T\mathbf{V}_{22} + h_{t-1}(v)^T\mathbf{U}_{22} \end{bmatrix}\right) \tag{19}$$

where $\tau$ is the learnable time constant; $\beta$ is the learnable scaling factor; $\Phi_\theta(s_t)$ is the feature representation of state $s_t$; $h_{t-1}(v)$ is the previous hidden state; and $\kappa = 0.1$ is the coupling strength between rings.

The cross-coupling matrices ($\kappa\mathbf{V}12$, $\kappa\mathbf{V}21$, $\kappa\mathbf{U}12$, and $\kappa\mathbf{U}21$) maintain a circular topology similar to the individual rings. A neuron at a particular position in the first ring connects most strongly to the neuron at the corresponding position in the second ring, with connection strength decreasing based on circular distance. This structured cross-coupling preserves spatial alignment between the two action dimensions while allowing semi-independent operation through the learnable coupling factor $\kappa$.

The final output provides action-values for both action dimensions simultaneously, preserving the spatial relationships within each ring while allowing for weak coupling between the rings.

### A.4.3    EXTENSION TO N RING CONFIGURATIONS

The double ring implementation extends to $R$ rings through a block matrix structure, where each ring encodes a distinct action dimension. For $R$ rings, the architecture uses block matrices $V_{\text{multi}} \in \mathbb{R}^{RN \times RM}$ and $U_{\text{multi}} \in \mathbb{R}^{RN \times RN}$, with diagonal blocks preserving individual ring dynamics and off-diagonal blocks handling cross-ring interactions via coupling parameter $\kappa$, as seen in Section A.4.2. While computational complexity scales as $O(R^2)$, selective coupling between only related dimensions creates a sparse structure with effective $O(R)$ complexity. This makes the approach viable for complex action spaces where actions decompose into multiple semi-independent planes, such as games combining movement, combat, and resource management dimensions.

## A.5 Models and Environments Implementation

We provide implementation details for both our models and the tested environments. For model implementations, EffZeroRA was applied across the Atari benchmark suite, while BDQNRA-UA and DDQNRA were specifically implemented for Highway and Mario Bros. Table 2 details the configuration of action spaces and ring architectures for each environment. The environments required different ring configurations based on their control schemes, ranging from single-ring implementations for basic movement to double-ring setups for more complex action spaces that combine movement and specialised actions. Each ring's topology was designed to preserve the natural relationships between actions, with central inhibitory actions included where appropriate as "no action".

Table 2: Implementation details for ring attractor architectures across environments. The table shows the environment (Env); ring configuration (Ring); number of actions or continuous 1D action space (Actions); inhibitory neuron placed equidistant to other neurons for "no action" term (Inhib); whether uncertainty estimation is used (Uncert); the implemented model (Model); and type of Neural Network used (Type).

| Game | | Configuration | | | Implementation | |
|---|---|---|---|---|---|---|
| Environment | Ring | Actions | Inhib. | Uncert. | Model | Type |
| Highway | Single | Continuous | No | Yes | BDQNRA-UA | CTRNN |
| Mario Bros | Single | 8 | No | Yes | BDQNRA-UA | CTRNN |
| Highway | Single | 8 | No | No | DDQNRA | DL-RNN |
| Mario Bros | Single | 8 | No | No | DDQNRA | DL-RNN |
| Alien | Double | 18 | Yes | No | EffZeroRA | DL-RNN |
| Asterix | Single | 9 | Yes | No | EffZeroRA | DL-RNN |
| Bank Heist | Double | 18 | Yes | No | EffZeroRA | DL-RNN |
| BattleZone | Double | 18 | Yes | No | EffZeroRA | DL-RNN |
| Boxing | Double | 18 | Yes | No | EffZeroRA | DL-RNN |
| Chopper C. | Double | 18 | Yes | No | EffZeroRA | DL-RNN |
| Crazy Climber | Single | 9 | Yes | No | EffZeroRA | DL-RNN |
| Freeway | Double | 18 | Yes | No | EffZeroRA | DL-RNN |
| Frostbite | Double | 18 | Yes | No | EffZeroRA | DL-RNN |
| Gopher | Double | 18 | Yes | No | EffZeroRA | DL-RNN |
| Hero | Double | 18 | Yes | No | EffZeroRA | DL-RNN |
| Jamesbond | Double | 18 | Yes | No | EffZeroRA | DL-RNN |
| Kangaroo | Double | 18 | Yes | No | EffZeroRA | DL-RNN |
| Krull | Double | 18 | Yes | No | EffZeroRA | DL-RNN |
| Kung Fu M. | Double | 18 | Yes | No | EffZeroRA | DL-RNN |
| Ms Pacman | Single | 9 | Yes | No | EffZeroRA | DL-RNN |
| Private Eye | Double | 18 | Yes | No | EffZeroRA | DL-RNN |
| Road Runner | Double | 18 | Yes | No | EffZeroRA | DL-RNN |
| Seaquest | Double | 18 | Yes | No | EffZeroRA | DL-RNN |

