# OpenReview forum: "Spatial-aware decision-making with ring attractors in Reinforcement Learning systems"
_ICLR.cc/2025/Conference — Submitted to ICLR 2025_

### Official Review · Reviewer_3UD9 · 2024-11-03

**Soundness:** 2
**Presentation:** 2
**Contribution:** 3
**Rating:** 5
**Confidence:** 4

**Summary:**

This paper proposes a novel approach to reinforcement learning (RL) that incorporates ring attractors—models inspired by neural mechanisms that encode spatial information—into the action selection process. By organizing actions on a ring structure and decoding decisions through neural activity, the method aims to improve learning speed, prediction accuracy, and stability in deep RL. When evaluated on the Atari 100k benchmark, it reported a 53% performance increase over baseline models, particularly in games with spatial components.

**Strengths:**

It is interesting to consider using ring attractor structure for decision making in RL.

The paper is well organized.

**Weaknesses:**

In the first approach, the rationale for using action values and their variances as connection weights is not well established. For instance, is there any prior work that employs Q-values as connection weights within a sub-network? Additionally, how should cases be handled when the Q-value exhibits very low variance? In Equation 7, adding the variance in the denominator of the weight calculation can lead to instability or weight divergence when the variance is close to zero.

The second approach, as presented in Section 3.2: “DL-based ring attractor integrated into the RL agent” is not well motivated. From “RNNs perform well in modeling sequential data and in capturing temporal dependencies for decision making ” to state that “RNNs mirror ring attractors’ temporal dynamics, with their recurrent connections and flexible architecture emulating the interconnected nature of ring attractor neurons. ” The relationship between the two parts, if any, are hard to find.

Ungrounded claims.  “Spiking neural networks (SNNs) are employed for their biological plausibility and efficient temporal information processing, which closely mimic natural neuronal behavior.” In this work, there is no spiking neural network. The ring attractor used a rectified linear unit, which is a rate unit, rather than a spiking unit.  The model in this work is also not "biologically plausible model” (Line 122). Both of the above claims are misleading and inaccurate.

**Questions:**

How the \mu_a in Eq. 12 be used in the model? Is it the mean in the expression of x_n in Eq. 1? But it was denoted as \alpha_a in Eq. 7.
Could the authors explicitly explain the relationship between these variables and their usage throughout the model?

Is there ring attractor at all in the model? The units interact through Eq. (5-6) COULD develop a ring attractor, but not all system interact like this actually develop ring attractor. This work did not spend any effort to confirm the system actually develop a ring attractor when using Q value as the connection weight. Could the authors provide specific evidence or analyses that would demonstrate the emergence of a ring attractor in their system? This can be achieved through visualizations or metrics that confirm the presence of ring attractor dynamics, as demonstrated in previous studies, such as Seung (1996, PNAS) and Kim (2017, Science). Although the 2017 Science paper is cited in this work, it is important to note that the concept of ring attractors in the brain has a longer history and a rich literature, tracing back to the 1996 PNAS study.

---

> ### Author Response · Authors · 2024-11-23
>
> We sincerely thank the reviewer for their thorough examination of our work's fundamental principles:
>
> **1 Q-values and variance lower bound**
>
> We thank the reviewer for their careful reading of our work. However, we believe there may be a misunderstanding about how Q-values are used in our model. The Q-values are not used as connection weights within the ring attractor network. Instead, the connection weights between neurons are fixed and defined by distance-dependent functions (w_(E_m→E_n) = e^(-d^2(m,n))), which define the fundamental topology and dynamics of the ring attractor.
> Q-values are used only as input signals (scaling factors K_i) to the ring attractor network, as shown in equation 7. This formulation allows learned action values to influence the ring attractor's activity pattern while maintaining its core structural properties. Regarding potential numerical instability from variance terms, it only has relevance at the initialisation point, as the uncertainty estimates from our Bayesian approach maintain reasonable lower bounds due to the inherent uncertainty in value estimation.
>
> **2 DL RNN-based implementation**
>
> The relationship between RNNs and ring attractors is fundamentally grounded in their shared ability to model continuous-time neural dynamics, as established by [Beer (1995)](https://journals.sagepub.com/doi/10.1177/105971239500300405)  in their work on CTRNNs. As demonstrated in our exogenous model based on Touretzky's ring attractor, these networks inherently rely on recurrent connections to maintain stable attractor states, providing a natural motivation for using RNNs in our Deep Learning implementation. Additionally, we are working on an appendix section to visualise the emergence of sustained ring patterns in our experiments.
>
> **3 SNN and biological plausibility**
>
> As highlighted by the reviewer, SNN is not relevant to this approach. We apologise for the misunderstanding and have corrected the methodology. Furthermore, we acknowledge ring attractors primarily serve heading direction representation, as shown by [Kim et al. (2017)](https://www.science.org/doi/10.1126/science.aal4835). Our innovation lies in adapting spatial awareness mechanisms for decision-making: just as biological ring attractors encode spatial representations for navigation, our model uses similar principles to organise actions in a circular topology.
>
> **Q1:** The symbol $\alpha$ represents the preferred orientation for a given action $a$, while $\mu_a$ represents the value associated with that action. For clarity, we have revised this notation since it was inconsistent with Equation 7, using now the same variable in both equations with the distinction that this one represents an average over a sum of samples $\bar{Q}(s,a)$.
>
> **Q2:** The symbol $\alpha$ represents the preferred orientation for a given action $a$, while $\mu_a$ represents the value associated with that action. For clarity, we have revised this notation since it was inconsistent with Equation 7, using now the same variable in both equations with the distinction that this one represents an average over a sum of samples $\bar{Q}(s,a)$.
>
> **Q3:** While the reviewer raises important points about verification of ring attractor dynamics, our model demonstrates key ring attractor properties inherent in its design and implementation:
> The spatial encoding of actions in our circular topology is not arbitrary, we enforce distance-dependent weighting through equations (5-6) that explicitly model local excitation and global inhibition, a remark of ring attractor dynamics.
> The performance improvements we observe in spatially-structured tasks (Table 1) suggest the model successfully maintains stable representations of spatial relationships between actions, consistent with ring attractor behaviour.
> Our ablation study (Section A.1.2) shows that randomly disrupting the ring structure significantly degrades performance, indicating the ring topology actively contributes to information processing rather than being merely architectural.
> We acknowledge and are working on additional visualisations of neural activity patterns that would strengthen our claims.
>
> Several neural architectures in machine learning have drawn inspiration from biological principles. One common example is Convolutional Neural Networks, which were influenced by the hierarchical organisation observed in the mammalian visual system [Hubel and Wiesel, 1962](https://pmc.ncbi.nlm.nih.gov/articles/PMC1359523/). While our research addresses a different domain, we similarly attempt to incorporate insights from neural mechanisms, though we acknowledge the substantial abstraction involved in translating biological principles to computational frameworks.
>
> We agree with the reviewer and moved [Zhang (1996)](https://www.jneurosci.org/content/16/6/2112) key proposal from the appendix to line 30 of the main text, reflecting its importance as first milestone.

---

> > ### Author Response · Authors · 2024-11-26
> >
> > We have addressed the reviewer's concern and expanded our work with new experimental results and supplementary appendices. See Manuscript Update at the top of the page for all changes made. We thank the reviewer for their thorough examination and encourage them to explore these additions.

---

> ### Comment · Reviewer_3UD9 · 2024-11-28
>
> The authors have partially addressed my questions and improved the presentation of the results in the revised manuscript. I would like to increase my score from 3 to 5.
>
> However, my question, "Is there a ring attractor at all in the model?" was not adequately addressed. I believe this is an important point that requires further clarification.

---

> > ### Author Response · Authors · 2024-11-28
> >
> > We appreciate the reviewer's feedback and for acknowledging the improvements in our revised manuscript. We especially appreciate raising a key question regarding whether our model truly resembles ring attractor structure and behaviour in the DL implementation. We can condense this evidence for our DL agent through several key aspects:
> >
> > **Structural Properties:**
> > As detailed in Section 3.2 and Appendix A.3, our model implements fundamental ring attractor dynamics through structured connectivity. The forward pass (V(s)) and hidden state (U(v)) are computed following distance-dependent weight functions defined in Equation 13, where both input-to-hidden connections and recurrent connections are laid out in circular topology. The learnable time constant τ controls information integration into the ring attractor as in CTRNN-based approaches. This architecture allows us to regulate input signals to the RNN layer, balancing spatial relationships with task-specific learning while preserving ring attractor dynamics.
> >
> > **Empirical Validation:**
> > As presented in our new appendix section A.2.3, we observe evidence of ring attractor dynamics preservation in our model, even when all parameter weights for the ring connections are set to be trainable. This means they can be evolved to what the DL algorithm expects to be the most efficient connectivity. As shown in Appendix A.2.3, the forward pass connections preserve the ring structure over training time, with distance-dependent decay patterns maintained throughout the learning process. This may suggest that the network naturally favours maintaining spatial topology for transmitting sensory information on a per-frame basis. The hidden-to-hidden connections demonstrate markedly different behaviour, evolving beyond their initial ring structure to develop specialised patterns that enable the encoding of environment-specific relationships between neurons in the hidden space.
> >
> > **Ablation Evidence:**
> > Our ablation studies in Section A.2.2 provide support for the relevance of the ring attractor structure. For the DL implementation, we removed the circular weight distribution to assess its importance. The significant performance degradation observed in the presented environment (Ms Pacman and Chopper Command) emphasises that the ring topology actively contributes to information processing rather than being merely architectural.

---

### Official Review · Reviewer_96JC · 2024-11-04

**Soundness:** 2
**Presentation:** 2
**Contribution:** 2
**Rating:** 6
**Confidence:** 3

**Summary:**

The authors propose incorporating ring attractors, a biologically-inspired neural circuit model for spatial information encoding, into reinforcement learning (RL) agents to improve action selection, particularly in spatially structured environments.
The authors explore two ring-attractor implementations on RL tasks: first, an exogenous spiking neural network (SNN) and second, a regular RNN based ring attractor.
They report significant performance improvements over baselines on the Atari 100k benchmark.

**Strengths:**

# Strengths
* It was interesting to see how a neural architecture found in biology was integrated into a machine learning setting. Physical ring attractors are found in biological organisms, such as the Drosophila fruitfly, and offer a strong inductive bias that could be used to increase sample efficiency as was done in this paper. (Note reservations below)
* Incorporating uncertainty in addition to tracking the mean is a nice contribution, potentially leading to more robust and adaptive behavior.
(Note reservations below)

**Weaknesses:**

# Weaknesses
* It was not clear at all to me why Spiking Neural Networks need to be involved in this paper. Regular "rate coding" RNNs should be enough to encode a ring-attractor. To the best of my knowledge, ring attractors in biology encode internal/cognitive/state variables, and aren't directly encoding action spaces. This discrepancy raises concerns about the biological relevance of the proposed approach.
* More clarity is needed on why the baselines chosen are justified. I think a better baseline would be to use a continuous 1-D circular variable (e.g. encoded as <sin \theta, cos \theta>) as the action space and then discretize it to match the action-space of the environment. This would be a more appropriate baseline to isolate the benefits of the ring attractor architecture itself.
* The paper doesn't fully explain the role and impact of uncertainty quantification. Maybe the authors should be exploring a simpler ring-attractor model which doesn't include uncertainty quantification.

**Questions:**

# Additional suggestions for improvement
* L135: "spatial exploitation"?
* L193: what are m and n here?
* L265: what is algorithm referring to here in "function approximation algorithm"?
* L343: typo: abs(m - m)
* L377: could you clarify how one neuron corresponding to one (s,a) still makes this a ring-attractor
* L406: typo: cummulative
* L408: How was \pi/6 chosen?
* L417: Can you clarify where this comes from -- "mean computational overhead ...."?
* L460: Can you provide at least one example each for how a game's action space has been mapped to Single and Double configurations
* The paper could use an overhaul in its organization. e.g. the ablation tests are essential to justify/clarify how the ring-attractor helps.

Related recent research worth citing
* Kutschireiter et al, "Bayesian inference in ring attractor networks", PNAS 2023
* Singh et al, "Emergent behaviour and neural dynamics in artificial agents tracking odour plumes", Nature Machine Intelligence 2023
* RI Wilson, "Neural Networks for Navigation: From Connections to Computations", Annual Review of Neuroscience 2023
* Xiong and Sridhar, "Understanding the Influence of Uncertainty and Noise on Spatial Decision Dynamics", Conference on Cognitive Computational Neuroscience 2024

Note that I did not spend significant time on the supplement

12/3: Increased my score from 5 to 6 for the authors' efforts revising the manuscript and responding to my feedback.

---

> ### Author Response · Authors · 2024-11-23
>
> We thank the reviewer for the valuable review; we especially appreciate the insights provided to expand on the experimental setup and validation context.
>
> **1 SNN and Biological Relevance**
>
> As pointed out by the reviewer, SNN is not relevant on this research, we apologise for the confusion. We've corrected the methodology to address this error. We use continuous-time recurrent neural networks (CTRNN) as the initial framework for the exogenous model.
>
> We appreciate the reviewer's perspective on ring attractors, and would like to offer a complementary interpretation based on spatial encoding evidence. [Kim et al. (Science, 2019)](https://www.science.org/doi/abs/10.1126/science.aal4835#:~:text=Ring%20attractors%20are%20a%20class,the%20representation%20of%20heading%20direction). demonstrated that ring attractors represent spatial awareness through heading direction, a neural code that bridges internal representation and spatial behaviour. Their work showed how bump-like activity patterns maintain a persistent sense of orientation while smoothly updating with the fly's movements. This continuous integration of spatial state and movement suggests that ring attractors have evolved to handle representations that are simultaneously internal (maintaining spatial awareness) and behaviorally relevant (guiding navigation).
>
> **2 Baselines and 1-D circular variable action space**
>
> The reviewer raises a very valid point that was considered and implemented by the authors. In our experiments, we mapped the highway's benchmark navigation action space as a one-dimensional circular variable. We recognise this was not explicitly mentioned. We are working to include this as a new section in the appendix, alongside information about other benchmark action spaces, ring attractor experimental layouts (both single and double configurations), and baseline-RA experimental integration.
>
> **3 Role and impact of uncertainty quantification**
>
> The role of uncertainty quantification (UQ) in our ring attractor model is fundamental to both its theoretical foundation and practical performance for the exogenous ring attractor model. Our experimental results in Figure 2 clearly demonstrate that the uncertainty-aware version (BDQNRA-UA) consistently outperforms both the baseline (BDQN) and the simpler ring attractor model without uncertainty (BDQNRA), providing empirical justification for its inclusion.
>
> The paper provides an explanation of UQ's role through mathematical formulations in Section 3.1.3 and explicit equations showing how uncertainty values (σₐ) directly influence the Gaussian functions driving ring attractor dynamics. This integration aligns with biological insights, as shown by recent work [Kutschireiter et al., 2023](https://pubmed.ncbi.nlm.nih.gov/36812206/), demonstrating the utility and performance of integrating uncertainty into ring attractors.
>
> Rather than adding unnecessary complexity, UQ is tightly integrated with the ring attractor architecture through the Gaussian activation functions (σᵢ = σₐ in Equation 7), creating a natural mechanism for uncertainty-aware action selection.
>
> **Q1:** Fixed, thank you.
>
> **Q2:** Here, $m$ and $n$ represent the positions (indices) of neurons within the ring network, where these indices define each neuron's location and determine the distance between any two neurons via $|m-n|$.
>
> **Q3:** A function approximation algorithm in this context is a method (typically a neural network) that learns to estimate Q-values (expected future rewards) instead of storing exact values for every possible state-action pair in reinforcement learning. It works by converting input states into feature vectors and using learned weights to approximate Q-values through the equation $Q(s,a) = θᵀx(s)$, making it practical for large or continuous state spaces.
>
> **Q4:** Fixed, thank you.
>
> **Q5:** While each neuron corresponds to an action-value pair, the ring attractor properties emerge from the circular connectivity pattern between neurons (defined by d(m,n) = min(|m - n|, N - |m - n|)) rather than from what the neurons represent.
>
> **Q6:** Fixed, thank you.
>
> **Q7:** Added brief explanation (line 410).
>
> **Q8:** In this context, "mean computational overhead" refers to the extra processing time or computational cost added by the ring attractor implementation. Whenstated"297.3% overhead", it means the integrated CTRNN ring attractor model (BDQNRA) took 3 times more time to run than the baseline model (BDQN).
>
> **Q9:** We are also to provide an appendix section that provides insights on integration for the different models and environments, including single and double ring implementations.
>
> **Q10:** We agree with reviewer in that ablation studies are essential to demonstrate the ring attractor's impact on learning performance across experiments. We are exploring a potential reorganization of the manuscript, though we are still determining which sections could be moved to the appendix given significant space constraints.

---

> > ### Comment · Reviewer_96JC · 2024-11-24
> >
> > Thanks to the authors for taking the time to respond to my feedback.
> >
> > 1: Re: "We appreciate the reviewer's perspective on ring attractors, and would like to offer a complementary interpretation based on spatial encoding evidence... ...demonstrated that ring attractors represent spatial awareness through heading direction... "
> >
> > Actually, this is exactly what I am saying, heading direction is an "internal/cognitive/state variable" and isn't directly encoding an action space. There are downstream computations that combine the internal heading angle variable and other internal variables (e.g. current goal heading) to generate an action (with RAs not playing a direct role).
> >
> > 2: "Baselines and 1-D circular variable action space" -- Thank you, this is a critical baseline. Looking forward to seeing it.
> >
> > Thank you for addressing the other comments!

---

> > > ### Author Response · Authors · 2024-11-26
> > >
> > > We appreciate the reviewer's insightful comments and would like to provide additional clarification.
> > >
> > > Q1: We completely agree with the reviewer's perspective on ring attractors. To clarify our position: while the ring attractor is indeed key for representing internal heading direction as a cognitive state variable, it is not working in isolation in this research as RL agent. Rather, it functions as part of an integrated system where the RL agent combines the ring attractor's output with other computational elements, provided by the baseline models to generate appropriate actions. We acknowledge that the ring attractor itself doesn't directly perform action selection in isolation, but rather contributes providing spatial encoding that, when combined with other components of the RL agent, enables more effective overall performance.
> > >
> > > Q2: We have submitted a revised version of our work that addresses several points, particularly incorporating the requested baseline comparisons for the 1-D circular variable action space. We invite the reviewer to examine these new additions, which we believe may better demonstrate the system's capabilities.

---

> > > > ### Comment · Reviewer_96JC · 2024-11-28
> > > >
> > > > Can you point me to where the additional material / changes relating to the baseline "incorporating the requested baseline comparisons for the 1-D circular variable action space" were made? I did not see it while skimming the text highlighted in orange in the revised manuscript.

---

> > > > > ### Author Response · Authors · 2024-11-28
> > > > >
> > > > > The new experiment, which evaluates BDQN, BDQNRA, and BDWNRA-UA models using a 1-D circular variable action space in the Highway environment, is presented in Figure 2, Section 4.1.
> > > > >
> > > > > Apologies for the confusion, to clarity, orange highlighting denotes only corrections to the original text and does not apply to extended content added to improve its readability.  Neither the new experiment nor the additional appendices (A.2.3, A.4, and A.5) are highlighted in orange.

---

### Official Review · Reviewer_xjv3 · 2024-11-04

**Soundness:** 2
**Presentation:** 2
**Contribution:** 3
**Rating:** 6
**Confidence:** 3

**Summary:**

This paper proposed a method for using a ring attractor mode as the final layer of a model-free reinforcement learning model on an agent with discrete actions for tasks in a 2D space. Given an observation and all actions, with the ring attractor, the Q value estimation can take advantage of prior knowledge of space sense for a better action decision. The experiments report that three models with this method outperformed the original baseline models in 2D video games.

**Strengths:**

This work contributes to an important concept in decision-making during reinforcement learning in a task with space: how to associate values with directions. This paper presents a method to do it with a ring attractor which plays a role in Q value fusion with a prior of space.

**Weaknesses:**

To my understanding, this method implicitly assumed that each action is associated with a direction, the Q value of different actions is a Gaussian distribution in different directions, and Q values in different directions can be summed up. These assumptions, which are not always true, were not discussed clearly in the paper. For example, given a task, not all actions are associated with a direction. In this case, optimising the ring attractor with gradient decent could lead to the degeneracy of dynamics. This paper should show if the ring attractor still works as expected after training.

The paper also did not present ring attractor models correctly, which contains fundamental errors. In section 3.1, the title claims that the ring attractor model being presented is a spiking neural network, however, the presented model is a firing-rate model without spikes. The equations describing the model look like an inappropriate combination of two different models because the definitions are not consistent. For example,  $i_n$ in equation (4) and (5) are different.  There are many other mathematical errors in the paper, although some of them are just typos, but impact the quality and soundness of the paper.

In the experiments, three different models that were not clearly reviewed or mentioned are presented. The name abbreviation of the models appears abruptly, thus a reader has to guess what they are, even the models this paper tried to propose are in this case. For example, what is BDQNRA, BDQNRA-UA, and EffZeroRA? And how they come from? Although the basic method to combine a network with the ring attractor is presented, the specific models as a result of the method should be introduced before presenting the results.

**Questions:**

1. Line 30. The ring attractor model was proposed much earlier than 2017, the original paper should be cited.
2. Line 128, 131, 169, 416. None of the models presented in this paper are spiking.
3. The content in Section 3.1.1 is not the contribution of the work. It should be in the related work or background.
4. In Lines 159 and 160, $i$ was referred to as an input signal, however, it is used as an index in later paragraphs. They are very different concepts.
5. Equation (2)(3). A mixture of differential and difference equations, especially "$u_{+\Delta t}$" is a confusing term.
6. Line 178, should $\Delta t$ be a constant in the context of ODE?
7. Equation (2)(3). What is $i_n$ given that there is only one inhibitory neuron? The equation in Line 193 about the weights from the inhibitory neuron to excitatory neurons is confusing.
8. Equation (4)(5) results different definitions of $i_n$.
9. Equation (6). Define $w^{I \rightarrow I}$.
10. Equation(7). Define $\alpha$ when it is not a function.
11. Equation(8). Wrong parentheses.
12. Line 249. The equation in this line contains a standalone parenthesis.
13. Equation(9). Why there is a normalisation?
14. Line 266 introduced $\Phi$ as an algorithm, but used as a function in later equations. What is the output of the function?
15. Line 264, 265 and 268, why $Q(s,a)=\Phi_\theta(s)=\theta ^ T x(s)$? Prove it or cite a reference.
16. Line 286 and 295, does $w_a$ represent the weights or the outputs of the upstream model?
17. Equation (11). What are $y$, $x'$ and $\Phi_{\theta_{target}}$?
18. Equation (13). Does the equation assume there are more than one inhibitory neurons? It is different from the early sections. $abs$ is in Italic thus not a good format to be a function. $d(m,n)$ defined twice and differently here. Why does the second definition of  $d(m,n)$ contain a term $(m-m$?
19. Table 1. What does the double ring mean? It is not explained mathematically.

**Details Of Ethics Concerns:**

The work does not raise any ethical concerns for me.

---

> ### Author Response · Authors · 2024-11-22
>
> We sincerely thank the reviewer for their insightful feedback, which has particularly strengthened our methodology:
>
> **1 Actions space assumptions**
>
> While our model does map actions to ring positions, as described in Section 3.1.2, it's not limited to purely directional actions. The paper demonstrates this through double ring configurations in Section 4.3, where games like Seaquest and BattleZone effectively use separate rings for movement and independent action dimensions. This architecture directly addresses cases where not all actions have inherent directional relationships.
> Regarding the potential for degeneracy in cases where actions lack directional mapping, Section 3.2 of our paper details how the DL implementation specifically handles this concern. The architecture employs a fixed forward path (V(s)) that maintains structural relationships through distance-dependent weights, while also incorporating trainable hidden state dynamics (U(v)) that can adapt to non-spatial action relationships.
> The empirical validation in Table 1 demonstrates that our approach works effectively across different action space configurations - from single ring games like Asterix (showing 110% improvement) to double ring games like Seaquest (32%). These results suggest that the ring attractor dynamics remain stable and effective after training, even when handling diverse action types. Additionally, ablation studies in the appendix indicate minimal performance degradation, compared to baseline, when actions are misplaced in a ring layout.
>
> **2 SNN and mathematical presentation**
>
> We apologise for the inconsistencies in the methodology, and we are working towards fixing all the issues that have been very well pointed out in this review. You are correct that Section 3.1 incorrectly describes a spiking neural network(SNN). We've corrected the methodology to address this error. We use continuous-time recurrent neural networks (CTRNN) as the initial framework for the exogenous model. Regarding the inconsistency between equations (4) and (5), we are working on reconciling all mathematical definitions and ensuring consistency across equations.
>
> **3 Experiment models presentation**
>
> We have added a brief clarification in line 380. Additionally, we will include integration specifications and implementation details for each model in the appendix. This revision should help readers better follow the relationship between our methodology and the specific model implementations discussed in the experimental results.
>
> **Q1:** We acknowledged [Zhang (1996)](https://www.jneurosci.org/content/16/6/2112), who first proposed ring attractors as a theoretical model for head direction cells, in the background placed in the appendix. As it is a key milestone we’ve moved the citation to line 30 in the text.
>
> **Q2:** Apologies for the confusion with the nature of the ring attractor model. We've corrected the methodology to address this error.
>
> **Q3:** Though these equations are well-known, their placement in the methodology is useful as they form the working components of our RL policy behaviour policy implementation. It may disconnect the theoretical foundation from our actual algorithmic steps, making the methodology harder to follow and reproduce.
>
> **Q4:** Acknowledged and working on it.
>
> **Q5:** Acknowledged and working on it.
>
> **Q6:** Acknowledged and working on it.
>
> **Q7:** Acknowledged and working on it.
>
> **Q8:** Acknowledged and working on it.
>
> **Q9:** Acknowledged and working on it.
>
> **Q10:** Acknowledged and working on it.
>
> **Q11:** Fixed, equations 5 and 6 wrongly displayed the current excitatory and inhibitory terms $v_n,u$ over the time integration constant $\tau$, which is incorrect. We have moved them outside of the equation parenthesis.
>
> **Q12:** Fixed, thank you.
>
> **Q13:**  Acknowledged and working on it.
>
> **Q14:**  Acknowledged and working on it.
>
> **Q15:** Fixed, added reference (line 268).
>
> **Q16:** They represent the weights of the upstream Bayesian Linear Regression model (line 287, 304).
>
> **Q17:** Fixed, developped further (lines 300-303).
>
> **Q18:** Fixed (equation13).In the DL RNN implementation, no inhibitory neurons are present. Instead, standard DL neurons with tanh activation maintain the attractor state through their hidden state dynamics, rather than through biological-like lateral inhibition. We are working to provide visualisation of the RNN forward and hidden state dynamics in a appendix section.
>
> **Q19:** The action space is split into two separate rings with weak connections between them, this implementation is not explicit in the original paper. We are also working to provide an appendix section that provides insights on integration for the different models and environments, including single anddouble ring implementations.

---

> > ### Comment · Reviewer_xjv3 · 2024-11-23
> >
> > The reply addressed some of my questions, but several points still require further clarification, and some others are acknowledged but not answered yet.
> >
> > 1. How does the RNN implementation of CANN differ fundamentally from the original model? Does the RNN version exhibit intrinsic dynamics? If intrinsic dynamics are present, it would be beneficial to include a figure illustrating these dynamics, such as curves of neuron activities under varying inputs.
> >
> > 2. Could the authors provide an explicit explanation of why normalisation is applied in Equation (9) in the comment?
> >
> > Please reply to the remaining questions if the authors make progress in the revision.

---

> > > ### Author Response · Authors · 2024-11-23
> > >
> > > We would like to thank the reviewer for the thoughtful review, particularly regarding CANN vs RNN intrinsic dynamics:
> > >
> > > **RNN vs CANN (CTRNN) intrinsic dynamics**
> > >
> > > We acknowledge the reviewer's important question about visualising the dynamics of our RNN-based ring attractor implementation. The fundamental challenge stems from the discrete, synchronous update nature of standard DL frameworks, making it difficult to replicate the continuous-time dynamics of biological ring attractors. While our RNN implementation incorporates key architectural elements - a learnable time constant τ and distance-based weight matrices V(s) and U(v) that capture spatial organisation (Eq. 13); it operates fundamentally in discrete time steps during both training and inference.
> > > To address this limitation and provide deeper insights into our model's dynamics, we are developing and will add a new appendix section. This section will present analyses of network behaviour through visualisation of: (1) emergence and stability of ring-shaped connection in the forward pass when fix weights constraints are removed for V(s), (2) temporal evolution of hidden states h(v) during action selection, (3) neuron activation patterns under varying input conditions, (4) stability analysis of attractor states by perturbing initial conditions, and (5) empirical investigation of how the learned time constant τ modulates information flow between input signals and hidden states.
> > > These visualisations, while constrained by the discrete nature of DL  frameworks, will provide concrete evidence for how our RNN implementation preserves essential computational properties of ring attractors. The analysis will complement our performance results from Section 4.3, offering insights into why our approach achieves significant improvements across diverse environments in the Atari 100k benchmark. This addition should also highlight promising directions for future research in approximating continuous-time dynamics within DL architectures.
> > >
> > > **Equation 9 normalisation and discrete action spaces**
> > >
> > > The action selection mechanism in Equation (9) presents an initial formulation of the mapping between ring attractor dynamics and reinforcement learning action selection. This basic mapping operates under two fundamental assumptions about the action space: discreteness and uniform distribution across the ring attractor's circumference. The normalization term N(A)/N(E) enables the transformation from neural activity to action selection by mapping from the higher-dimensional neural representation (N(E) excitatory neurons) to the lower-dimensional discrete action space (N(A) actions). Given the uniform distribution assumption, each action corresponds to a contiguous arc of neurons in the ring, with the scaling factor ensuring that the maximally activated region of neural activity (identified by argmax_n{V}) maps to the appropriate discrete action index. This formulation maintains the ring attractor's spatial encoding properties while accommodating the discrete nature of the action space. A comprehensive treatment of various integration approaches, including single and double ring configurations and continuous action spaces, along with specific environment implementations, will be added in a forthcoming appendix section that is currently work in progress.

---

> > > > ### Author Response · Authors · 2024-11-24
> > > >
> > > > We would like to express our gratitude again to the reviewer. We have addressed all their comments, which have improved the quality of our work:
> > > >
> > > > **Q1**: We acknowledged Zhang (1996), who first proposed ring attractors as a theoretical model for head direction cells, in the background placed in the appendix. As it is a key milestone we've moved the citation to line 30 in the text.
> > > >
> > > > **Q2**: Apologies for the confusion with the nature of the ring attractor model. We've corrected the methodology to address this error.
> > > >
> > > > **Q3**: Though these equations are well-known, their placement in the methodology is useful as they form the working components of our RL policy behaviour policy implementation. It may disconnect the theoretical foundation from our actual algorithmic steps, making the methodology harder to follow and reproduce.
> > > >
> > > > **Q4**: We agree with the reviewer that $i$ refers to index, being $x$ the input signals to the ring attractor. We have clarified this (lines 155,161). Additionally we have changed notation for excitatory and inhibitory connections from $e, i$ to $\epsilon, \eta$ to address the same potential issue.
> > > >
> > > > **Q5**: To avoid confusion, we keep the ordinary notation and drop the middle fraction term. We aim to convey through equation 2,3 that we approximate a differential equations with a discrete difference equation.
> > > >
> > > > **Q6**: Yes, you are absolutely right. It is a constant in the context in ODE, states in line 179. We could drop the derivative equation but it provides context of the approximation we are performing. $\tau$ as constant does not depend anymore in time, hence the approximation we are showing in eq 3.
> > > >
> > > > **Q7**: The reviewer is absolutely correct, elaborating on weighted connection $i_n$ (named now to avoid confusion with indexes $\eta_n$) the suffix $n$ alludes to excitatory neuron $n$ it connects. We've addressed equation in 193 to explicitly show the result when only one inhibitory connection is placed in the ring.
> > > >
> > > > **Q8**: This has been fixed in equations 4,5, thank you.
> > > >
> > > > **Q9**: $w^{I \rightarrow I}$ was not strictly needed as it is a recurrent connection with distance 0 for the inhibitory neuron results in redundant weight with value 1 ($e^0$). We've removed the term in equation 6.
> > > >
> > > > **Q10**: Fixed, thank you. We have defined $\alpha_n$ (line 230) and unified the naming convention for $\alpha_a(a)$.
> > > >
> > > > **Q11**: Fixed, equations 5 and 6 wrongly displayed the current excitatory and inhibitory terms $v_n,u$ over the time integration constant $\tau$, which is incorrect. We have moved them outside of the equation parenthesis.
> > > >
> > > > **Q12**: Fixed, thank you.
> > > >
> > > > **Q13**: The normalization term N(A)/N(E) enables the transformation from neural activity to action selection by mapping from the higher-dimensional neural representation (N(E) excitatory neurons) to the lower-dimensional discrete action space (N(A) actions). Given the uniform distribution assumption, each action corresponds to a contiguous arc of neurons in the ring, with the scaling factor ensuring that the maximally activated region of neural activity (identified by argmax_n{V}) maps to the appropriate discrete action index. This formulation maintains the ring attractor's spatial encoding properties while accommodating the discrete nature of the action space. A comprehensive treatment of various integration approaches, including single and double ring configurations and continuous action spaces, along with specific environment implementations, will be added in a forthcoming appendix section that is currently work in progress.
> > > >
> > > > **Q14**: We agree with the reviewer. To clarify notation, the terminology here $\Phi_\theta(s)$ represents both a function approximation algorithm and a function, common in ML [LeCun et al., 1998](http://vision.stanford.edu/cs598_spring07/papers/Lecun98.pdf). It represents the algorithmic process of approximating Q-values through neural network weights ($\theta$) and feature extraction, while mathematically functioning as a mapping from states to Q-values through $\theta^T x(s)$. The output of the function/ function approximation algorithm are the Q-values for a state-action pairs.
> > > >
> > > > **Q15**: Fixed, added reference (line 268).
> > > >
> > > > **Q16**: They represent the weights of the upstream Bayesian Linear Regression model (line 287, 304).
> > > >
> > > > **Q17**: Fixed, developed further (lines 300-303).
> > > >
> > > > **Q18**: Fixed (equation13). In the DL RNN implementation, no inhibitory neurons are present. Instead, standard DL neurons with tanh activation maintain the attractor state through their hidden state dynamics, rather than through biological-like lateral inhibition.
> > > >
> > > > **Q19**: The action space is split into two separate rings with weak connections between them, this implementation is not explicit in the original paper. We are also working to provide an appendix section that provides insights on integration for the different models and environments, including single and double ring implementations.

---

> > > > > ### Comment · Reviewer_xjv3 · 2024-11-25
> > > > >
> > > > > The authors have responded to all my questions and they are mostly clear now, although there is a repetitive equitation in line 876. The rating is improved to 6.

---

> > > > > > ### Author Response · Authors · 2024-11-26
> > > > > >
> > > > > > We appreciate the reviewer taking the time to look at the appendix and pointing out inconsistencies, we are looking at the appendix equations and working on them to avoid repetition and improve clarity.

---

> > > > > > > ### Author Response · Authors · 2024-11-26
> > > > > > >
> > > > > > > We have addressed the reviewer's concern by removing redundant equations from line 1011 that did not add substantive context to the section. We have also expanded our work, see Manuscript Update at the top of the page, with new experimental results and supplementary appendices, which we encourage the reviewer to examine.

---

### Official Review · Reviewer_6Rpb · 2024-11-04

**Soundness:** 3
**Presentation:** 3
**Contribution:** 2
**Rating:** 5
**Confidence:** 4

**Summary:**

The authors propose adapting the ring attractor network, a long-studied, biologically inspired neural network architecture, as a stochastic action selection module for reinforcement learning. The authors demonstrate mathematically how this model can integrate input uncertainty and convert it to probabilistic attractor states, which could be used to control action choices. The authors further embed the ring network model with existing deep reinforcement learning models, including BDQN, DDQN and EfficientZero and show significant boost in performance on the Atari game suite.

**Strengths:**

The ring attractor model is a prevalent and important biological architecture with demonstrated fucntion in sensory integration and decision-making. Testing its utility as part of a deep reinforcement learning module is an interesting and promising direction. The mathematical formulation of the model is well presented. An extensive list of empirical evaluation has been done showing significant boost of performance from the EfficientZero baseline.

**Weaknesses:**

1)	Novelty: The ring attractor model itself has been proposed decades ago and many theoretical work has examined how this model architecture integrate information to output perceptual or action decisions. The ability of the ring attractor network to integrate or compute uncertainty has also been analyzed theoretically (e.g. Kutschireiter et al., PNAS 2023). So the novelty of the work does not lie on the theoretical analysis on the ring attractor model, but on its application as a action-selection module in RL. While the empirical evaluation appears promising, there is no strong theoretical insight on why this model excels over existing algorithms for action selection (see more details below).
2)	Insight on why the model excels at action sampling: The authors claim that the ring attractor-based action selection introduces uncertainty awareness (UA) into action selection. However, the BDQN algorithm, which served as the baseline for BDQN-UA, already performs uncertainty estimation and action selection through approximate Thompson-sampling. While the ring attractor model may bring additional level of stochasticity into action sampling, it is unclear what the key mechanism that brings about the boost in empirical performance. In fact, as shown in Kutschireiter et al. PNAS 2023, computation in the ring attractor network essentially implements Bayesian inference. Since Bayesian inference is already done through BLR in the BDQN model, the goal performing another round of inference through the ring network appears unnecessary.
3)	While EffZeroRA should remarkable performance on the Atari suite, it is unclear whether the performance boost arise due to uncertainty estimation through the BLR on action values, or through the deployment of the ring attractor. Perhaps an ablation study comparing EffZero with BLR/Thompson sampling against EffZeroRA could help isolate the role of the ring attractor
4)	Compatibility of ring structure to arbitrary action spaces: A feature in the dynamics of the ring attractor model is that nearby units tend to have correlated activity (due to strong excitatory connections). When used for action selection, this structure feature could introduce correlation in the sampling probability of actions represented by close-by units on the ring. However, this correlation in sampling probability may not be desirable for all RL tasks, hence hindering the generality of the ring attractor as an action selection module.
5)	Implementation details--could the authors comment on: 1. Are the ring weights fixed or updated during training? 2. If updated, how is the appropriate decay of excitation weights maintained? 3. Does DDQN-RA use BLR for uncertainty estimation like BDQN-RA? 4.What specific role does neuronal spiking play in the exogenous model, and is it necessary?

**Questions:**

Could the authors comment on:
1) mechanistic insights on why the ring attractor model adds a performance boost on top of BDQN, which already performs Bayesian uncertainty estimation and posterior sampling of actions?
2) whether the sampling implicit bias in the ring model (i.e. sampling probability of “close-by” actions may be correlated due to local excitation among ring units) could impact its generality in RL applications?
3) provide more implimentation details as listed in point 5 above.

---

> ### Author Response · Authors · 2024-11-22
>
> Thank you for this excellent feedback, it has given us meaningful ways to strengthen our work:
>
> **1 Novelty**
>
> Problem Statement
>
> The problem statement lies in conveying faster learning for decision-making algorithms, particularly within the RL framework. While ring attractors have been well-studied in neural models of decision-making, their theoretical integration into fundamental decision-making frameworks beyond neuroscience remains limited. We acknowledge that ring attractor models have been studied in theoretical work. However, our paper's novelty lies in several key contributions:
>
> Novel Integration with RL/DL
>
> We are the first to propose using ring attractors as a mechanism for action selection in RL, creating a bridge between neuroscience-inspired models and RL/DL architectures to provide spatial-aware decision-making that increases the learning rate of modern algorithms. To apply this concept to more mainstream approaches, we create a novel implementation of ring attractors using RNNs (Section 3.2),  compatible with DL frameworks, while preserving their key properties the Section  3.2.
>
> Our approach introduces explicit spatial encoding of actions, which is fundamentally different from existing methods. As demonstrated in Section 4.3, this leads to performance improvements, particularly in spatially-structured tasks (e.g., 110% improvement in Asterix, 105% in Boxing).
>
> Theoretical Insights
>
> Equations 5,6 describe the behaviour of our policy. We prove through ablation studies, Figure 4 and Figure 5 that the dynamics in the ring and correct layout of the action space as input signals is the only way that yields actual learning improvements.
>
> **2 Insight on why the model excels at action sampling**
>
> Ring attractors enhance BDQN's performance by encoding actions in a circular topology while incorporating uncertainty through Gaussian variance parameters σₐ, mathematically expressed through input signal equations $x_n(Q) = \sum_{a=1}^{A} {\frac{Q(s,a)}{\sqrt{2\pi \sigma_a}} \exp\left(-\frac{1}{2}\frac{(\alpha_n - \alpha_a)^2}{\sigma_a^2}\right)}$. This structure of the action space provides benefits beyond BDQN's Bayesian estimation by using explicit spatial encoding in the behaviuor policy plus uncertainty quantification expressed as variance of the action signals. As demonstrated by [Sun et al. (2018)](https://www.researchgate.net/publication/326227083_An_Analysis_of_a_Ring_Attractor_Model_for_Cue_Integration), ring attractors provide a robust framework for combining cues according to their certainty, using this to our advantage during action selection.
>
>
> **3 Unclear performance boost between EffZeroRA and uncertainty estimation**
>
> Atari implementation uses our DL-based ring attractor model without uncertainty estimation. Uncertainty is only used in the CTRNN exogenous model with BDQN's BLR layer.
>
> **4 Compatibility of ring structure to arbitrary action spaces**
>
> As described in Section 3.1.2, actions are mapped to specific locations on the ring based on their spatial relationships, providing significant flexibility that enables generalisation. Our approach has demonstrated robust performance across diverse action spaces in the Atari 100k benchmark. Single-ring configurations excel in directional movement games (Asterix, Ms Pacman), while double-ring configurations effectively handle games combining movement with independent action dimensions (Seaquest, BattleZone). Additionally, ablation studies in the appendix indicate minimal performance degradation, compared to baseline, when actions are misplaced in a ring layout.
>
> **5 Implementation details**
>
> 1. This is n clarified in section 3.2.1 (line 349)
> 2. Neural Ring Dynamics (Section 3.2.1): The learnable time constant τ controls information integration into the ring attractor. The forward pass uses fixed distance-dependent weights to maintain spatial relationships, while the hidden state dynamics U(v)m,n employs trainable weights to learn complex action dependencies. By adjusting τ, we can regulate input signals to the RNN layer, balancing spatial relationships with task-specific learning while preserving ring attractor dynamics. We are working on a new appendix section demonstrates the emergence of sustained ring patterns in our experiments.
> 3. It does not, all DL-based implementations are implemented as outlined in **3 Unclear performance boost between EffZeroRA and uncertainty estimation**.
> 4. SNN does not play any role in this research, we apologise for the confusion. We've corrected the methodology to address this error. We use continuous-time recurrent neural networks (CTRNN) as the initial framework for the exogenous model.
>
> **Q1**
>
> This has been developed in the section above: **2 Insight on why the model excels at action sampling**.
>
> **Q2**
>
> This has been developed in the section above: **4 Compatibility of ring structure to arbitrary action spaces**.
>
> **Q3**
>
> This has been developed in the section above: **5 Implementation details**.

---

> > ### Author Response · Authors · 2024-11-26
> >
> > We would like to let the reviewer know we have provided an appendix section (A.2.3) showing the emergence of sustained ring patterns in our experiments. We have also expanded our work with new experimental results and other supplementary appendices, see Manuscript Update at the top of the page for all changes made. We appreciate the reviewer's requests that helped develop the presentation of the key concepts in this research.

---

> > > ### Comment · Reviewer_6Rpb · 2024-12-03
> > >
> > > I appreciate the authors’ effort in addressing several common concerns raised by me and other reviewers. I have just a few remaining questions and would love to hear from the authors:
> > >
> > > **1. Role of the ring attractor in uncertainty quantification (UQ):** The authors mention UQ as one of the advantages of the RA model. However, in the BDQN-UA framework (which is the only model that embodies uncertainty estimation), the uncertainty $\sigma_a\$ is quantified through BDQN and then directly substituted into eq. 1. Then what role does the RA play in UQ?
> > >
> > > **2. What is the nature of the spatial relationship encoded by RA? And why does it improve action sampling even in the absence of UQ?** Based on eq. 1 & 13, nearby neurons in the ring share similar or correlated inputs and strengthen each other’s activity through local excitation. If nearby neurons are used to output values for actions targeting nearby locations, it would result in positive correlations in action values that are not necessarily desirable. For example, moving left and up in an Atari game may have quite different values and it is unclear a priori why one would want to enforce correlation among them. Could the authors elaborate on when such spatial correlation in action values are desirable? Furthermore, I noticed a few Atari games that were implemented in the original EffZero paper was not tested here (incl. Amidar, Assault, Demon Attack). Could the authors comment on why their action space may not be compatible with the RA?
> > >
> > > **3. I appreciate the newly added Appendices, though am still baffled at some implementation details.** E.g. 1) In line 349-351 the authors describe the input-to-hidden weights as fixed, though the first equation in eq. 13 contains the learnable parameter $\lambda$, which also appears in the hidden-to-hidden weight. 2) in line 404, the authors mention that \sigma_a is held fixed at $\pi/6$ in BDQNRA, which “enables smooth action transition while preventing interference with opposing actions”—how does one determine this value for an arbitrary task with a different action space? 3) In eq. 19 of Appendix A. 4.1., is $Q(s,a)$ computed in one step or does it involve multiple inference steps as illustrated in Fig. 8?
> > >
> > > **4.  To help clarify some of the questions above, would the authors be open to share to their code?**

---

> ### Author Response · Authors · 2024-12-03
>
> We appreciate the reviewer's insightful comments and would like to provide further insights:
>
> **1. Role of the ring attractor in uncertainty quantification (UQ):**
> The ring attractor primarily serves as a spatial encoding mechanism, while uncertainty quantification happens through the BDQN framework. The RA architecture provides a structured topology for organizing actions and their uncertainties, but does not directly participate in computing those uncertainties. The variance estimates from BDQN's Bayesian calculations feed into the Gaussian input signals to the ring attractor (via σₐ in eq. 1), allowing uncertainty to influence the activation patterns, and influencing the action selection process. The correct distribution and injection of uncertainty measurements into the ring improves action selection as presented in Figure 2 in Section 4.1 of the experiments and supported further by the ablation study in Appendix A.2.1. Ring attractors enhance BDQN's performance by encoding actions in a circular topology while incorporating uncertainty through Gaussian variance parameters σₐ, mathematically expressed through input signal equations $x_n(Q) = \sum_{a=1}^{A} {\frac{Q(s,a)}{\sqrt{2\pi \sigma_a}} \exp\left(-\frac{1}{2}\frac{(\alpha_n - \alpha_a)^2}{\sigma_a^2}\right)}$. This structure of the action space provides benefits beyond BDQN's Bayesian estimation by using explicit spatial encoding in the behavior policy plus uncertainty quantification expressed as variance of the action signals. As demonstrated by [Sun et al. (2018)](https://www.researchgate.net/publication/326227083_An_Analysis_of_a_Ring_Attractor_Model_for_Cue_Integration), ring attractors provide a robust framework for combining cues according to their certainty, using this to our advantage during action selection.
>
> **2. What is the nature of the spatial relationship encoded by RA?** The spatial organisation provided by ring attractors fundamentally enhances action selection by explicitly encoding the action space and enabling the distribution of spatial representations across the neural network. The correlation between actions is solved by a trainable hidden space as presented in the appendix, Sections A.2.3, A.4; and methodology, Section 3.2.1. As detailed in Section 4.2, games with primarily directional movement like Asterix utilise a single-ring configuration for eight directional movements, while games combining movement with independent actions like Seaquest employ a double-ring configuration; one for movement and another for secondary mechanics. The biological plausibility of this approach can be traced back to the studies presented in the appendix, Section A.1, where ring attractors provide spatial cue integration that we employ in the context of action selection.
>
> **3. I appreciate the newly added Appendices, though am still baffled at some implementation details.**
> The learnable time constant τ in equation 13 controls the rate of information integration, since the weights are fixed to preserve the ring topology this learnable parameter enables that rate of transfer from previous layers in the DL agents can vary depending on the preference of the DL agent. As detailed in Section 3.2.1, this allows the network to balance spatial relationships with task-specific learning while preserving ring attractor dynamics through the fixed distance-dependent connection weights.
> The selection of σₐ = π/6 for BDQNRA was empirically determined to provide optimal balance between action discrimination and smooth transitions in the tested environments. As shown in Section 4.1, this value enables smooth action transitions while preventing interference with opposing actions. This middle step (BDQRA) in the integration between BDQN and a full integration with uncertainty with BDWNRA-UA was put in place to make easier the distinction on how each of the components spatial representation and uncertainty was actually providing an improvement compared to baseline.
> Equation 19 in Appendix A.4.1 is computed in a single forward pass during inference, despite Fig. 8 illustrating the temporal evolution of hidden states for visualization purposes. The complete forward pass combines input signals and hidden state information in one step through the matrix operations defined in the equation.
>
> **4. To help clarify some of the questions above, would the authors be open to share to their code?** We acknowledge the importance of sharing our codebase for future research and are actively working on preparing it for release. While we were initially planning to share this at a later stage, we understand the value of making it available sooner. We will have the implementation details ready by the end of this week, including baselines for BDQN, DDQN, and Efficient Zero. Although we weren't able to prepare the codebase within one day of the request, it will be available at [this repository](https://github.com/marcosaura/RA_RL).

---

### Author Response · Authors · 2024-11-26
**Manuscript update**

We are grateful to the reviewers for their insightful feedback which has helped us develop a more comprehensive manuscript. We believe these additions strengthen the paper's potential to spearhead new research bridging neuroscience-inspired models with standard DL approaches.
We are providing an updated manuscript that expands and clarifies several key sections:

– 4.1 EXOGENOUS RING ATTRACTOR MODEL PERFORMANCE ANALYSIS: We expand our analysis of the exogenous ring attractor model to include continuous action spaces, demonstrating its efficacy in mapping ring attractor outputs to a continuous 1D circular action space using the OpenAI Highway environment.

– A.2.3 DEEP LEARNING RING ATTRACTOR MODEL EVOLUTION: We introduce an examination of the ring attractor dynamics in our DL implementation. We analyse the evolution of ring-shaped patterns during training, revealing how the network naturally preserves spatial relationships while adapting to task-specific requirements.

– A.4 DEEP LEARNING RING ATTRACTOR MODEL IMPLEMENTATION DETAILS: We present implementation details covering both single and double-ring configurations. We provide the mathematical framework underlying different implementations and outline pathways for extending the approach to more complex configurations.

– A.5 MODELS AND ENVIRONMENTS IMPLEMENTATION: We offer systematic documentation of our experimental implementations, detailing the specific properties of each environment-model pairing. This addition includes configuration details and clarifies the particular requirements of different environments, to improve reproducibility and facilitating future applications of this approach.

---

### Meta-Review · Area_Chair_D925 · 2024-12-21

**Metareview:**

This paper proposes to use ring attractor network components into Q-learning based reinforcement learning. The general idea is to provide spatial information and relationships for actions (e.g. arrow-keys in video games) and induce correlations, rather than having RL agents learn actions as independent and separate choices.

The paper further claims that by adding ring attractors, this leads to uncertainty estimates over the Q-function, which allows better decision making over unseen environment areas.

Experiments are conducted over:
* Super Mario Bros (discrete action space) and Highway (Most likely discrete?)
* A subset of the Atari100K benchmark

where adding the ring attractor improves performance over base methods.

One core issue as an outsider to the ring attractor framework is that the paper has not been easy to follow, despite Figure 1 attempting to represent the mechanics. There is too much dense notation that makes the ring attractor's design difficult to imagine, and I think more effort is required to make it accessible and impactful to a wider audience.

Regarding other issues, I relied on the reviewer discussion (see below).

**Additional Comments On Reviewer Discussion:**

Post-rebuttal, this paper obtained an extremely balanced score between rejection and acceptance, i.e. (5,5,6,6). In the initial review cycle, the scores were lower (e.g. a 5 was previously a 3).

This was a personally difficult read for me, as I don't have enough background knowledge on ring attractors and their biological significance. Therefore I relied much more on the feedback of all the reviewers who do have such knowledge.

The main issues raised by reviewers around this topic have been:
  * There is no actual spiking neural network in the model / possible mathematical mischaracterizations of the network.
  * It's unclear what contributions the ring attractor component brings to uncertainty quantification, when the last layer weights of the Q-function are already made to be probabilistic / allow Bayesian linear regression
  * Whether a ring attractor is "inside the model at all?"

While the authors have attempted to resolve this issues during the rebuttal, there are still clarity issues which remain (as seen from e.g. Reviewer xjv3's 19 questions on details). Given the additional borderline scores, I overall recommend rejection for now - I think this paper requires additional polishing to be resubmitted to the next ML conference.

---

### Decision · Program_Chairs · 2025-01-22

Reject